# Synapse types are spatially associated with regional hemodynamics in the mouse brain

Justine Y. Hansen[1], Andrea I. Luppi[1,2], Zhen Qiu[3,4], Silvia Gini[5], Ben D. Fulcher[6], Alessandro Gozzi[5], Seth G. N. Grant[4], Bratislav Misic[1]*

1 Montréal Neurological Institute, McGill University, Montréal, Quebec, Canada, 2 Centre for Eudaimonia and Human Flourishing, Department of Psychiatry, University of Oxford, Oxford, United Kingdom, 3 Department of Biomedical Engineering, University of Strathclyde, Glasgow, United Kingdom, 4 Centre for Clinical Brain Sciences, University of Edinburgh, Edinburgh, United Kingdom, 5 Center for Neuroscience and Cognitive Systems, Istituto Italiano di Tecnologia, Rovereto, Italy, 6 School of Physics, The University of Sydney, Sydney, New South Wales, Australia

* bratislav.misic@mcgill.ca

## Abstract

Synapses are the connections that transform neurons from simple electrically charged cells into complex circuits that support perception, cognition and action. Recent advances in single-punctum synapse mapping in mice make studying synapse diversity and differential expression possible. How do diverse synapses relate to the spatial patterning of whole-brain dynamics? Here we map the spatial distribution of multiple synapse types to >6 000 time-series features from fMRI recordings in awake mice to understand the comprehensive dynamical phenotype of multiple synapse types. We find that specific synapse types are associated with specific features of haemodynamics, including high-amplitude events and signal stationarity. These variations in synapse types and dynamics are associated with the structural and functional network embedding of brain regions. Finally, using two additional fMRI datasets in anaesthetized mice, we show that synaptic protein lifetime reflects differential synaptic engagement across behavioural states. Collectively, this work suggests that the spatial organization of microscale synapse types may shape whole-brain dynamics.

## Introduction

The brain is composed of billions of interconnected neurons that support the propagation of electrical signals and the emergence of cognition and behaviour. A fundamental unit in all neural circuits is the synapse. While hodology is typically focused on whether a synapse exists (i.e., whether two neurons are connected), a frequently overlooked question is what type of synapse is present, and how this synaptic phenotype shapes its function. Indeed, different excitatory synapses express different neurotransmitter receptors, ion channels, and signaling complexes [1,4,5]. This

**Data availability statement:** All code and data used to conduct the analyses are available at https://github.com/netneurolab/hansen_synaptome and on Zenodo (doi: 10.5281/zenodo.18201390).

**Funding:** BM acknowledges support from the Natural Sciences and Engineering Research Council of Canada (RGPIN-2017-04265), Canadian Institutes of Health Research (PJT-180439), and Canada Research Chairs Program (CRC-2022-00169). JYH acknowledges support from the Helmholtz International BigBrain Analytics & Learning Laboratory and the Natural Sciences and Engineering Research Council of Canada. AG acknowledges support by the European Research Council (ERC-DISCONN No. 802371, BRAINAM-1224 ICS No. 101125054) and Sara and Paolo Baracchino. The funders had no role in study design, data collection and analysis, decision to publish or preparation of the manuscript.

**Competing interests:** The authors have declared that no competing interests exist.

cocktail of synaptic proteins shapes the voltage dynamics of a single neuron [6–8]. In other words, brain function emerges not only from complex patterns of neural wiring, but also from the specific synaptic phenotype at every individual point of contact. Incorporating the brain's synaptic heterogeneity is necessary for understanding how regional synaptic architecture ultimately shapes regional brain dynamics and connectivity.

Recent single-punctum synapse mapping technology in mice has made it possible to image the proteomic composition of individual synapses throughout the whole brain, revealing an immense diversity of synapse types—the synaptome [1,9]. Two synaptic proteins in particular are well-suited candidates for genetic tagging in mouse models: postsynaptic density 95 (PSD95) and synapse-associated protein 102 (SAP102). These two postsynaptic scaffolding proteins are stably and abundantly expressed in excitatory synapses, assemble receptors, channels, and other signaling molecules into multiprotein signaling complexes, and play a direct role in shaping the synapse's response to a neural signal [1,6,8]. PSD95 and SAP102 synapses can be further classified according to morphological features into multiple synapse subtypes, each with unique properties. For example, some synapse subtypes stably express PSD95 over multiple weeks ("long-lifetime" synapses), while other synapse subtypes recycle PSD95 within a matter of hours ("short-lifetime" synapses), a possible mechanism underlying memory, cognitive flexibility, and learning [2]. Furthermore, mice that do not express PSD95 or SAP102 demonstrate abnormal synaptic transmission, as well as cognitive and learning deficits that are specific to the type of synapse that is affected [6–8,10]. Importantly, synapses that express only PSD95 versus those that express only SAP102 are differentially expressed in the brain, such that each brain region has its own unique synaptic composition or synaptome architecture [1,9].

The regional heterogeneity of synaptome architecture suggests that different brain regions are equipped to generate different patterns of neural dynamics. Indeed, whole-brain recordings of neural dynamics in mice consistently demonstrate spatial variation in local spontaneous activity [11,12]. These patterns of dynamics are highly structured: they can be organized into networks of distributed areas with similar function [13–16], and change depending on behavioural state [17,18]. Furthermore, recent data-driven feature extraction software has made it possible to comprehensively describe, using thousands of statistical time-series features, a rich dynamical phenotype of every brain region [16,19,20]. What then are the mechanistic origins of these regionally heterogeneous dynamical phenotypes?

In the present report, we ask whether the dynamical phenotype of a brain region can be traced back to its underlying synaptic phenotype. Using whole-brain quantifications of PSD95- and SAP102-expressing synapses in a single mouse brain, alongside awake resting-state functional magnetic resonance imaging (fMRI) recordings in a separate population of 10 mice, we compare the spatial distribution of synapse types to > 6 000 features of regional fMRI time-series. After establishing that each synapse type is associated with specific dynamics, we test whether a region's synaptic profile influences its embedding in global structural and functional networks. We then ask whether the functional role of a synapse type is ubiquitous across awake

and anaesthetized behavioural states, or whether some synapses play a larger role during wakefulness. Finally, in multiple sensitivity and robustness analyses, we (1) validate synapse density maps in an independent dataset, (2) confirm our findings are not driven by fMRI signal-to-noise ratio (SNR), (3) confirm signal amplitude and time-series features do not reflect in-scanner motion, and (4) confirm our findings are not driven by the spatial localization of different cell types. Throughout, we find that the differential expression of synapse types maps onto unique features of macroscale haemodynamics and interregional connectivity.

## Results

Approximately one billion individual synaptic puncta were imaged in a single whole mouse brain, using fluorescent markers for two synaptic proteins: PSD95 and SAP102 [1]. Morphological features (e.g. size, shape) of all PSD95-exclusive, SAP102-exclusive, and PSD95/SAP102 colocalized synapses were quantified and clustered into 11 subtypes of PSD95 synapses, 7 subtypes of SAP102 synapses, and 19 subtypes of colocalized synapses. Synapse density for these 37 synapse subtypes are mapped to 775 regions of the Allen Reference Atlas (ARA) [23] (Fig 1A). By clustering the synapse density similarity matrix, defined as the pairwise Spearman correlation of synapse subtype expression across brain regions, we find that all SAP102 synapse subtypes are similarly spatially expressed (Fig 1B). For PSD95 synapses, we find two clusters of expression profiles that align nearly perfectly with the recycling rate of PSD95 within the synapse—that is, PSD95 lifetime (Fig 1B barplot; see *Methods* for details on lifetime quantification; note that lifetime quantification was not performed for SAP102) [2]. We therefore separately analyze long-lifetime PSD95 synapses (6 subtypes) and short-lifetime PSD95 synapses (5 subtypes). Finally, we find that PSD95/SAP102 colocalized synapses follow the expression pattern of either SAP102-exclusive synapses ("SAP102-like colocalized synapses") or long-lifetime PSD95-exclusive synapses ("PSD95-like colocalized synapses") (S1 Fig). We therefore focus on long-lifetime PSD95 and SAP102 synapses in the main text, and show remaining synapse subtypes in the supplement.

To integrate the synapse density data with structural (tract-tracing) and functional (fMRI) datasets, we map the 775 samples to the regions included in the structural (137 right hemisphere regions) and functional (88 bilateral regions) data (Figs 1C, 1D, and S2). For each synapse type—long-lifetime PSD95, short-lifetime PSD95, and SAP102—we show their average spatial expression profile on an axial, coronal, and sagittal view of the mouse brain (Fig 1E and 1G; see S1 Fig for colocalized synapse types). All synapse types are more populated in anterior regions such as the isocortex, cortical subplate, and olfactory areas. Long-lifetime PSD95 and SAP102 synapses demonstrate opposing dorsal-ventral density gradients in the coronal slice, with long-lifetime PSD95 synapses being more populated dorsally and SAP102 synapses being more populated ventrally. Short-lifetime PSD95 synapse density is highest in the medulla but otherwise relatively uniform across isocortical, olfactory, and subplate areas.

### Synapse types are associated with unique macroscale dynamics

At the level of an individual synapse, its molecular make up and morphology is associated with a unique physiological response [6,8], but does whole-brain synapse density map onto unique features of brain activity? To address this question, we analyze spontaneous fMRI activity in awake mice ($N = 10$), and compare regional variation in haemodynamics to regional variation in synapse types. Fig 2 shows example time-series from brain regions with progressively more long-lifetime PSD95 synapse density (Fig 2A) and SAP102 synapse density (Fig 2B; time-series at the bottom are from regions with the greatest density). Visually, we observe that these BOLD signals are qualitatively different (Fig 2A and 2B). Namely, time-series in regions with greater long-lifetime PSD95 synapse density appear less periodic and more variable than regions with lower long-lifetime PSD95 synapse density (Fig 2A). Likewise, time-series in regions with high SAP102 synapse density show high-amplitude events (Fig 2B).

While this initial visual inspection hints at potential relationships between synapse density and time-series properties, we turn to time-series analysis methods to comprehensively phenotype the dynamic signature of each brain region.

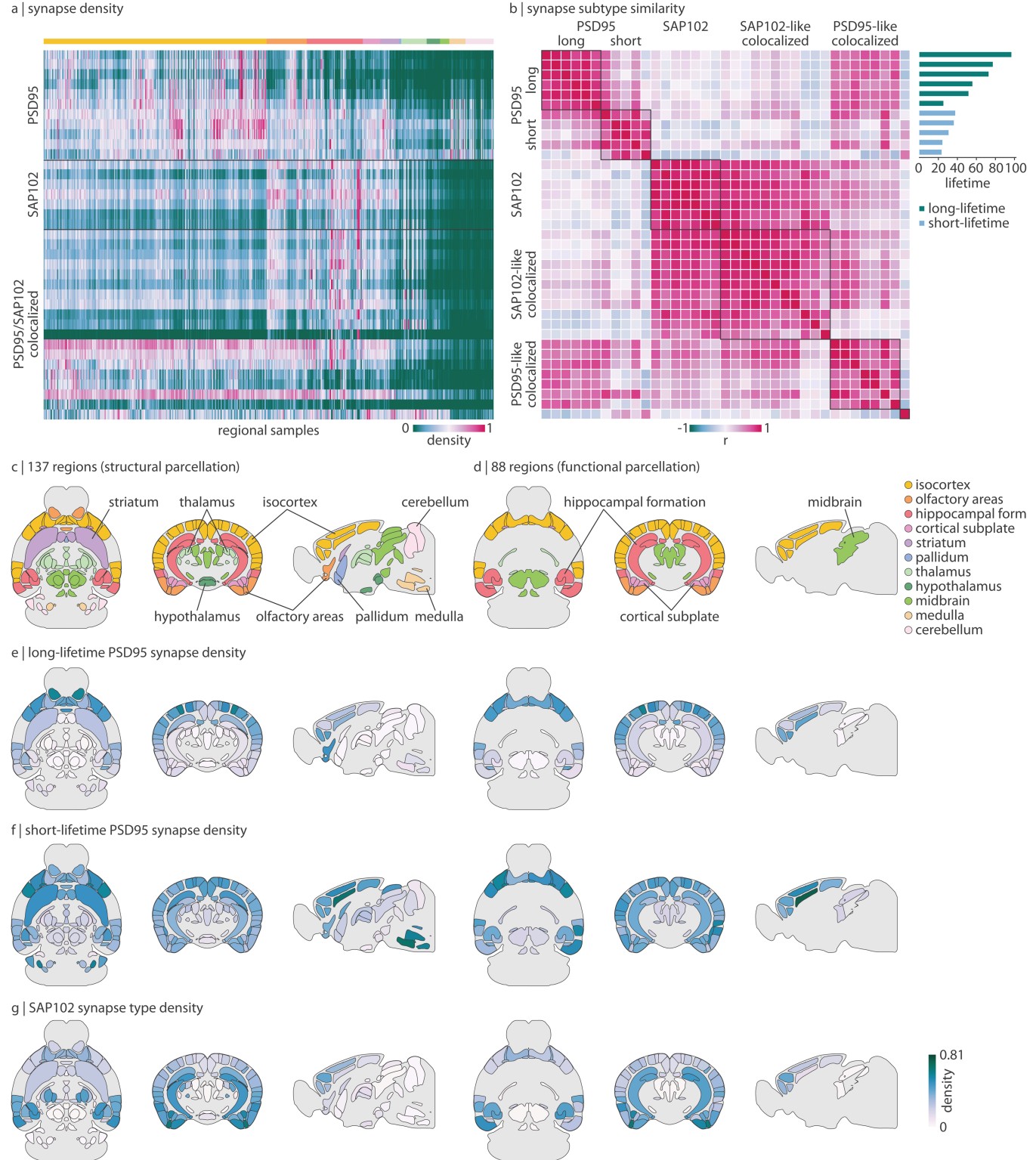

**Fig 1**. **Synapse type density in the mouse brain.** PSD95 and SAP102, two postsynaptic scaffolding proteins expressed at excitatory synapses, were fluorescently imaged at single-punctum resolution across the whole mouse brain [1]. (a) Normalized synapse density across 775 regional samples (ontological structure indicated by the horizontal bar and the legend in panel c–d) [1]. PSD95-expressing synapses, SAP102-expressing synapses, and

PSD95/SAP102 colocalized synapses are divided into subtypes (rows) according to morphological parameters. (b) Heatmap (symmetric): pairwise spatial correlation (Spearman's *r*) of synapse density. Black squares represent clusters identified using the Louvain community detection algorithm. Barplot: protein lifetime of PSD95 subtypes [2]. Green bars indicate long-lifetime PSD95 synapses and blue bars indicate short-lifetime PSD95 synapses. (c–d) Axial, coronal, and sagittal view of the 137-region parcellation used for analyses involving tract-tracing data (c) and the 88-region parcellation used for analyses involving fMRI data (d). Note that, since tract-tracing data is limited to the right hemisphere, the structural parcellation encompasses 137 right hemisphere regions. We show mirrored data on the left hemisphere. Regions are coloured and labeled according to ontological structure. (e–g) Axial, coronal, and sagittal view of mean long-lifetime PSD95 synapse density (e), mean short-lifetime PSD95 synapse density (f), and mean SAP102 synapse density (g), mapped to both 137- and 88-region parcellations. The colourbar in (g) applies to all three panels e–g. See S1 Fig for brain maps of colocalized synapse types. All mouse brains are plotted using `brainglobe-heatmap` [3].

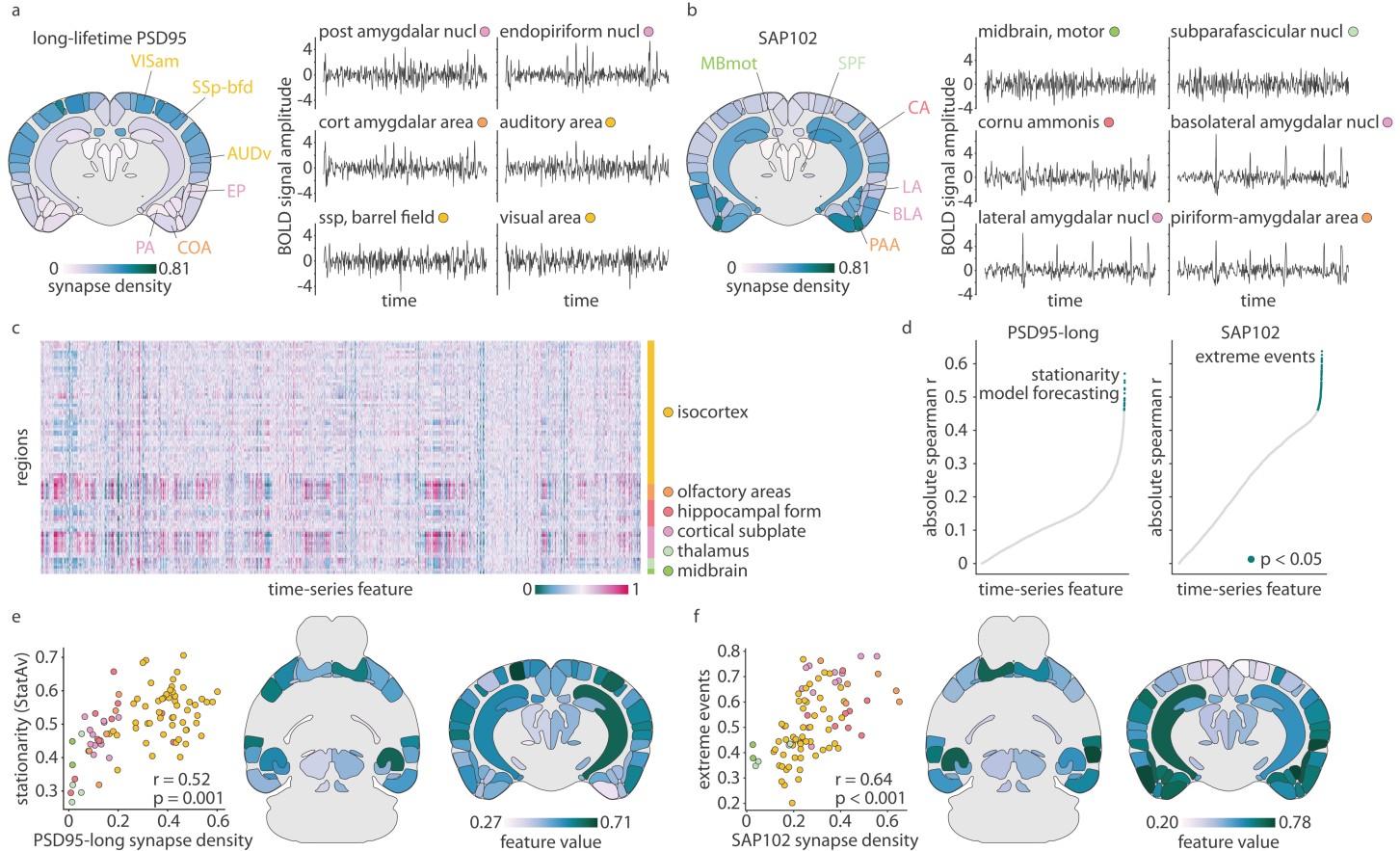

**Fig 2. Synapse types are associated with unique macroscale dynamics.** Resting-state awake fMRI data was acquired in 10 mice and time-series properties were correlated with synapse type densities. (a–b) Six example z-scored time-series from brain regions with variable long-lifetime PSD95 (a) and SAP102 (b) synapse density. Synapse density is shown on a coronal slice with the example brain regions labelled. Region labels on the coronal slice and coloured dot to the right of the brain region's name indicates the region's major ontological structure (legend in panel (c)). (c) Using `hctsa`, > 6 000 time-series features were calculated for each of 88 regional time-series in all mice. Time-series feature values were normalized to the unit interval, averaged across mice, and are shown in the heatmap. Brain regions are ordered according to ontological structure, as indicated by the vertical bar. (d) Each time-series feature was correlated (Spearman's r) with synapse density, for long-lifetime PSD95 and SAP102 synapses separately. Each point on the scatter plots represent a correlation coefficient, where green points indicate statistical significance after multiple comparisons correction (p<0.05, Bonferroni-corrected). Time-series features were sorted by correlation magnitude, and a selection of key, statistically-significant features are listed in the figure. (e) Scatter plot showing the correlation between long-lifetime PSD95 synapse density and `StatAv10`, a measure of stationarity (standard deviation of the mean value of ten equally-sized time-series segments; greater value indicates greater standard deviation so lower stationarity; S3 Fig [21]). (f) Scatter plot showing the correlation between SAP102 synapse density and the timing of extreme events in a time-series (S4 Fig). All mouse brains are plotted using `brainglobe-heatmap` [3].

Specifically, we use the highly comparative time-series analysis toolbox (`hctsa`) to compute 6 471 statistical features of each time-series in every region and mouse, including measurements of autocorrelation, entropy, frequency composition, signal amplitude distribution, and predictability [19,20]. Features are then averaged across mice (Fig 2C). Next, we correlate (Spearman's $r$) each synapse type with all 6 471 time-series features (Fig 2D). This generates a list of time-series features that are significantly correlated with each synapse type (Bonferroni-corrected; S1 Table). Within each list, many time-series features measure similar phenomena; we therefore select a representative time-series feature that best summarizes each list (S3A, S4A Figs; see *Methods* for details).

We find that long-lifetime PSD95 synapse density is most correlated with features related to signal stationarity, such that regions with more long-lifetime PSD95 synapses have lower stationarity and more variable signal. Indeed, the cluster of features that are most correlated with long-lifetime PSD95 synapse density can be summarized by a simple statistic called `StatAv` [21], which is defined as the standard deviation of $n$ time-series segment means (S3B Fig). Fig 2E shows the correlation between long-lifetime PSD95 synapse density and `StatAv10` (`StatAv` over 10 non-overlapping time-series segments; $r = 0.52$, Bonferroni-correted $p = 0.001$). Meanwhile, SAP102 synapse density is most correlated with features related to the presence and timing of extreme (outlier) events, such that regions with more SAP102 synapses tend to demonstrate more high-amplitude events, especially later in the time-series. In Fig 2F we show the correlation between SAP102 synapse density and a time-series feature that measures the median timing of extreme outlier events ($r = 0.64$, Bonferroni-corrected $p < 0.001$; see S4B Fig for a deeper intuition of this representative time-series feature). As expected, we confirm that these dynamical phenotypes extend to colocalized synapses (S5, S6 Figs). Lastly, while we do repeat these analyses for short-lifetime PSD95 synapses (S7 Fig), we find a poor alignment between synapse density and time-series features, suggesting that short-lifetime PSD95 synapses are not reliably associated with specific features of haemodynamics.

## Synapse types colocalize with structural and functional hubs

Do synapse types—with their unique dynamic properties—make distinct macroscale structural and functional connections? Using a dataset of axonal projections in the mouse brain (Allen Mouse Brain Connectivity Atlas [22]), we constructed a $137 \times 137$ weighted and directed structural connectivity matrix (Fig 3A). First we ask whether synapse density is correlated with structural weighted in-degree, that is, the sum of all afferent connection strengths into a single region. Since PSD95 and SAP102 are expressed postsynaptically, they are inherently markers of afferent connections. Indeed, despite their having distinct spatial expression profiles, we find that both long-lifetime PSD95 and SAP102 synapses are significantly positively correlated with weighted in-degree (long-lifetime PSD95: $r = 0.41$, $p < 0.001$; SAP102: $r = 0.42$, $p < 0.001$; Fig 3B, left). Meanwhile, only SAP102 synapses—that is, those that mark high-amplitude events in the time-series—are significantly correlated with weighted out-degree (sum of all efferent connection strengths; long-lifetime PSD95: $r = 0.09$, $p = 0.28$; SAP102: $r = 0.39$, $p \approx 0$; Fig 3B, right; see S8A Fig for similar findings in colocalized synapse types). Short-lifetime PSD95 synapses again show poor alignment between synapse density and degree (in-degree: $r = 0.18$, $p = 0.03$; out-degree: $r = -0.16$, $p = 0.07$). One possible explanation for this dichotomy between synapse type and relationship with anatomical connectivity is that only some synapses generate the necessary dynamics needed for the neuron to establish axonal connections.

Global anatomical connectivity is also correlated with SAP102 synapse density—and not correlated with PSD95 synapse density—when we consider layer-specific synapse density in the isocortex (maximum 31 regions). By correlating weighted in- and out-degree with layer-specific synaptic densities, we find that only layer-specific SAP102 synapse density is significantly correlated with structural degree. SAP102 synapse density in layer IV—the primary recipient of afferent connections—is significantly correlated with weighted in-degree ($r = 0.58$, $p = 0.023$) and SAP102 density in layer II/III—the main source of cortico-cortical efferent connections—is significantly correlated with weighted out-degree ($r = 0.36$, $p = 0.049$; Fig 3C). Furthermore, we find that the correlation between PSD95 synapse density and structural degree is closely

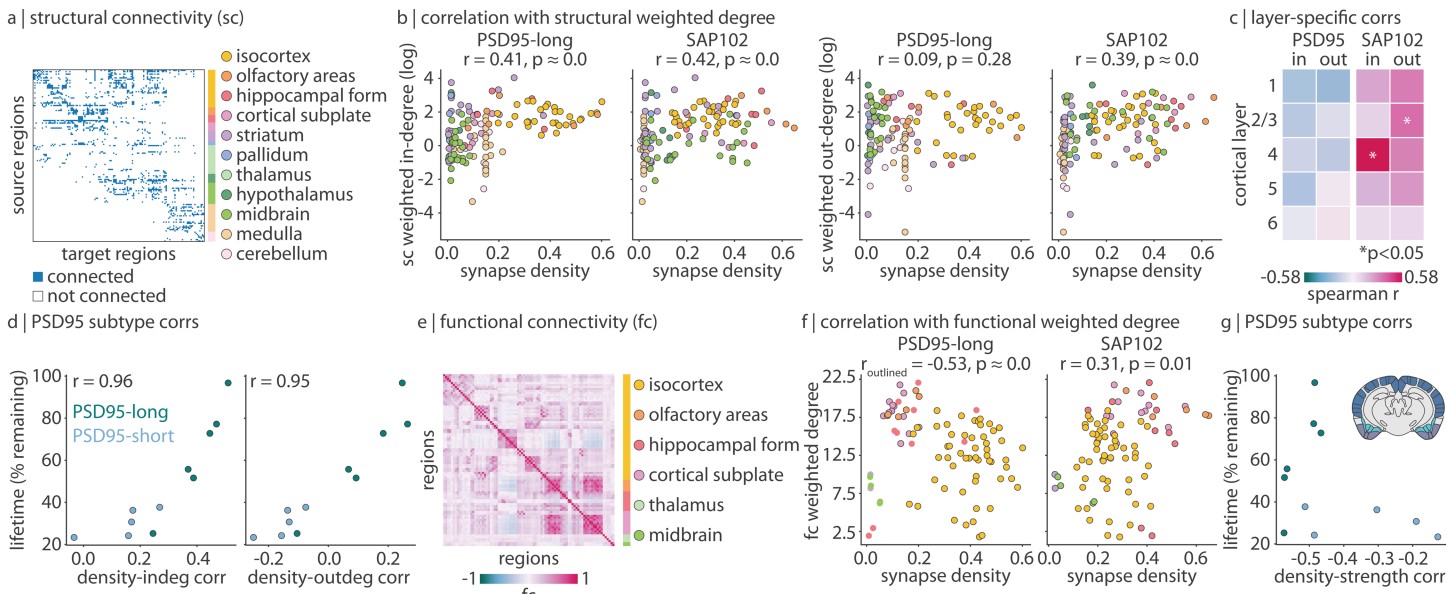

**Fig 3. Synapse types colocalize with structural and functional hubs.** (a) Structural connectivity matrix from tract-tracing data [22]. The matrix is binarized for visualization. (b) Correlations between long-lifetime PSD95 or SAP102 synapse density and weighted in- or out-degree of structural connectivity (sum of all afferent or efferent connections to or from a brain region, respectively). Weighted degree (*y*-axis) is log-transformed for visualization. Each point is a brain region, and regions are coloured according to their ontological structure. (c) Synapse density within each of 5 layers in the isocortex were separately correlated with weighted in- and out-degree. Asterisks represent *p* < 0.05. (d) Synapse density for each of 11 PSD95 synapse subtypes were correlated with weighted in- and out-degree of SC. Scatter plots show the density-degree correlation (Spearman's *r*) on the *x*-axis and PSD95 subtype lifetime on the *y*-axis. Each point represents a PSD95 subtype, and points are coloured according to whether they fall into the "long-lifetime" cluster or the "short-lifetime" cluster, as in Fig 1B. (e) Average functional connectivity (Pearson's *r* between pairs of regional time-series) across all mice. (f) Correlations between long-lifetime PSD95 or SAP102 synapse density and weighted degree of functional connectivity (sum of functional connectivity between one region and all other regions). Each point is a brain region, and regions are coloured according to their ontological structure. In the left-most scatter plot, only isocortical, olfactory, and cortical subplate regions are outlined and included in the correlation calculation. (g) Density-strength correlation (*x*-axis) versus synapse lifetime (*y*-axis) for all 11 PSD95 subtypes, as in panel (d). Note that the correlations are calculated using isocortical, olfactory, and cortical subplate regions only (ontological structures coloured in the coronal slice inset).

related to synapse lifetime: synapses with longer lifetime are more positively correlated with structural in- and out-degree (*r* = 0.96 for weighted in-degree, *r* = 0.95 for weighted out-degree; Fig 3D; see S8B Fig for similar findings in colocalized synapse types).

We next compare synapse density with functional connectivity (pairwise correlation between regional fMRI time-series). Specifically, we correlate synapse density with regional weighted degree of functional connectivity, a measure of functional integration (defined as the sum of functional connectivity strengths between one region and all others; Fig 3F). We find that short-lifetime PSD95 synapses are not correlated (*r* = −0.08, *p* = 0.46), and SAP102 synapses are weakly correlated (*r* = 0.31, *p* = 0.01), with functional weighted degree (see S8C Fig for similar findings in colocalized synapse types). On the other hand, when we visualize the correlation between long-lifetime PSD95 synapse density and functional weighted degree, we find a strong negative correlation within cortical (isocortical, olfactory, and cortical subplate) structures (*r* = −0.53, *p* ≈ 0), that is, when excluding structures with near-zero synapse density (e.g. midbrain, thalamus, part of hippocampus). In other words, regions with fewer (but non-zero) long-lifetime PSD95 synapses (and more stationary signal) tend to be functional hubs. Indeed, regions with many long-lifetime PSD95 synapses (and less stationary signal) are located in functionally specialized regions such as visual, somatosensory, and auditory cortex (Fig 1E). Again, when we consider the 11 PSD95 subtypes individually, we find that all long-lifetime PSD95 synapses are correlated at *r* ≈ −0.5 (−0.57 ≤ *r* ≤ −0.45) with functional weighted degree, whereas short-lifetime PSD95 synapse correlation coefficients range

from −0.52 to −0.14 (Fig 3G). Collectively, these findings demonstrate a dichotomy between long-lifetime PSD95 and SAP102 synapses: PSD95 synapses may be important for regional functional specialization, and SAP102 synapses may be involved in establishing a stable wiring architecture in the brain. In other words, the synaptic makeup of a region may shape its dynamical features and ultimately its embedding in large-scale networks.

## Synapse types mediate structure-function relationships

Given that structural and functional organization reflects synaptic architecture, we next ask whether synapse types mediate the coupling between brain structure and function [24,25]. We define regional structure-function coupling as the fit ($R^2_{adj}$) of a simple linear regression model that predicts a region's functional connectivity profile from its structural connectivity profile [26–29]. To bypass the limitation that structural connectivity is a sparse matrix, we calculate the communicability of the structural connectome, which results in a full matrix describing how easily information diffuses from one region to another. Next, we add either long-lifetime PSD95, short-lifetime PSD95, or SAP102 synapse density to the regression model and compare how model fit changes. This analysis is conducted using the 35 right hemisphere brain regions present in the structural, functional, and synapse density datasets. In Fig 4A we show scatter plots where each point represents a brain region, the *x*-axis represents structure-function coupling when using structural connectivity alone, and the *y*-axis represents structure-function coupling when synapse density is included in the model. As expected, most points, especially those representing non-isocortical regions, lie above the identity line (grey line in Fig 4A; see coronal slices in Fig 4A for change in coupling ($\Delta R^2_{adj} > 0$)).

Thus far, short-lifetime PSD95 synapses consistently demonstrate weak or no correlation with features of macroscale haemodynamics, anatomical connectivity, and functional connectivity (S7 Fig). These synapses may house proteins with too quick a recycling rate to establish long-term dynamics (that can be measured with fMRI) or connections. We hypothesize that these short-lifetime PSD95 synapses, in comparison to long-lifetime PSD95 and SAP102 synapses, may be more relevant for the cognitive flexibility required when awake versus anaesthetized. We therefore extend our analyses to a dataset of fMRI recordings acquired on mice anaesthesized with halothane ($N = 19$; see *Methods* for details). We ask: does short-lifetime PSD95 synapse density only improve structure-function coupling when mice are awake?

Specifically, we recompute structure-function coupling before and after the addition of the three synapse types using a functional connectivity matrix derived from anaesthetized mice (Fig 4B). We then compare how the addition of synapse type changes the regional distribution of $\Delta R^2_{adj}$ for awake versus anaesthetized mice (Fig 4C). For long-lifetime PSD95 and SAP102 synapses, structure-function coupling increases are not significantly different in anaesthetized versus awake mice (Wilcoxon signed-rank $p > 0.05$). Short-lifetime PSD95 synapses on the other hand demonstrate a significantly reduced $\Delta R^2_{adj}$ in anaesthetized mice, that is, short-lifetime PSD95 synapse density information improves structure-function coupling significantly more in awake versus anaesthetized mice (Wilcoxon signed-rank $p < 0.001$; for $\Delta R^2_{adj}$ stratified by functional system, see S9 Fig). For completeness, we include colocalized synapses in the synapse-informed model and find $\Delta R^2_{adj}$ is significantly increased (Wilcoxon signed-rank $p < 0.05$) in anaesthetized versus awake mice, suggesting first that colocalized synapses dominate the change in structure-function coupling, and second that colocalized synapses contribute a similar but stronger effect as long-lifetime PSD95 and SAP102 synapses (S10 Fig).

Finally, to ensure generalizability across different anaesthetics, we replicate these findings using a separate dataset of mice anaesthetized with a combination of medetomidine and isofluorane ($N = 14$, S11 Fig). Short-lifetime PSD95 synapses again significantly increase structure-function coupling in awake mice; however, the decrease in structure-function coupling in awake mice for long-lifetime PSD95 synapses and SAP102 synapses are now statistically significant. This change may reflect subtle differences in fMRI dynamics between the two anaesthetic states. For example, medetomidine/isofluorane has been shown to shift the spectral components of BOLD fluctuations towards higher frequencies [30], while halothane better preserves the spectral properties of BOLD fluctuations [31].

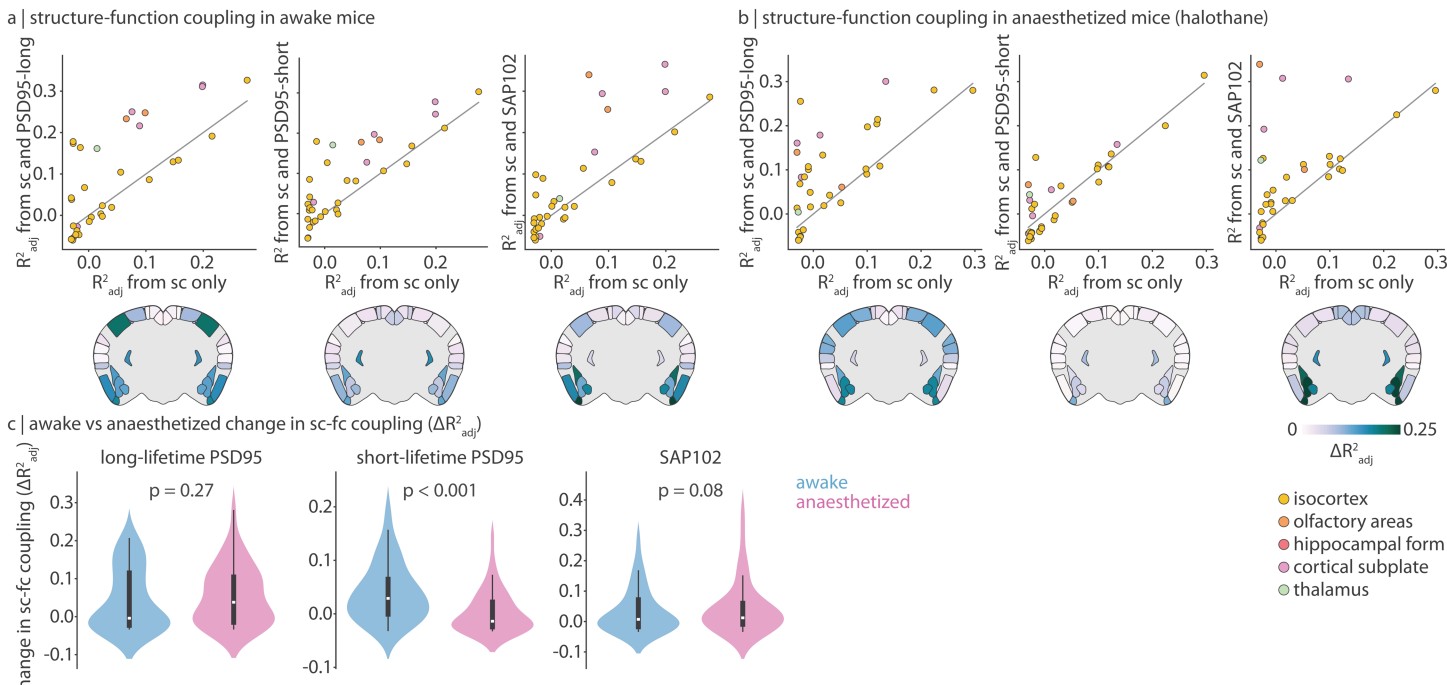

**Fig 4**. **Synapse type density improves structure-function coupling** For each brain region, we predict its functional connectivity profile from either regional communicability alone (a measure of structural connectedness) or regional communicability alongside synapse type density. Structure-function coupling is defined as the fit ($R^2_{adj}$) of the model. (a–b) Using functional connectivity from awake (a) or anaesthetized (with halothane; b) mice, we calculate structure-function coupling before (*x*-axis) and after (*y*-axis) adding synapse type density (long-lifetime PSD95, short-lifetime PSD95, or SAP102) to the model. Each point is a brain region, and points are coloured according to major ontological structure. The identity line is shown in grey. Coronal slices show the change in coupling ($\Delta R^2_{adj}$) after adding synapse density to the model. Data are mirrored across hemispheres for visualization. (c) We compare the change in structure-function coupling (distribution of $\Delta R^2_{adj}$) in awake versus anaesthetized mice, for all three synapse types separately. Statistical significance is assessed using a two-sided dependent non-parametric t-test for paired samples (Wilcoxon signed-rank test). Violin plots estimate a kernel density on the underlying data, the white point represents the median, the thick vertical line represents the quartiles of the distribution, and the thin vertical line represents the range. Data underlying violinplots can be found at https://github.com/netneurolab/hansen_synaptome/blob/main/results/violinplot_underlying_data.xlsx. All mouse brains are plotted using `brainglobe-heatmap` [3].

## Sensitivity and robustness analysis

In this final subsection, we conduct six analyses to gauge the sensitivity and robustness of the current findings: (1) we confirm synapse density data is coexpressed with synapse-related gene expression data from a larger population of mice, (2) we confirm our findings are not driven by fMRI SNR, (3) we ensure fMRI signal amplitude and time-series features are not driven by motion, (4) we confirm our findings remain consistent when global signal regression is applied to the fMRI signal, (5) we assess the replicability of regional time-series phenotypes and spatial time-series feature value distributions, and (6) we confirm that these results are specific to synapse types as opposed to spatial variation in cell types.

First, PSD95 and SAP102 synapse density are derived from a single male mouse, raising the concern that these synapse density maps are specific to the individual and are not generalizable to a broader population. Due to technological limitations, high spatial resolution synapse density data is not available for a larger population of animals. While this hinders our ability to validate synapse density maps, we instead apply gene expression data from in situ hybridization experiments in the Allen Mouse Brain Atlas to test whether synapses measured in a single mouse are coexpressed with synaptic genes measured across multiple mice [23]. We correlate synapse type density with expression levels of 19 919 genes across 275 unique brain regions. Next, we summarize the biological processes of genes that are most coexpressed

with each synapse type using a Gene Ontology analysis (see *Methods* for details). Despite long-lifetime PSD95 and SAP102 synapses following largely independent expression patterns ($r = 0.22$ in this 275-region parcellation), all three demonstrate greatest coexpression with genes involved in synaptic organization, plasticity, and development (Fig 5A and 5B, S12A, S13 Figs).

Furthermore, among the 1 295 most stably and reproducibly expressed genes, we find that all synapse types are again highly coexpressed with synaptic genes (Figs 5C, 5D, and S12B Fig). For example, long-lifetime PSD95 synapse density is highly correlated with genes such as *Dact2* (involved in neural development and synapse formation; $r = 0.79$), *Slc9a9* (regulates ion balance across intracellular membranes; $r = 0.76$), *Lamp5* (regulates inhibitory synaptic transmission; $r = 0.73$), and multiple *Kcn* genes (potassium channel subunit gene s, e.g. *Kcnq5*: $r = 0.73$). Likewise, SAP102 synapses are highly expressed with genes encoding synaptic proteins such as *Akap12* (scaffolding protein involved in signal transduction; $r = -0.69$), *Kcng4* and *Kctd9* (voltage-gated potassium channels; $r = -0.64$, $r = -0.63$, respectively), *Cacna2d2* (voltage-gated calcium channel; $r = -0.64$), and *Ddn* (Dendrin, a postsynaptic protein thought to be involved in retrograde signaling).

Second, we ensure that the relationship between synapse density and dynamics isn't driven by fMRI SNR. SNR was calculated as the mean signal within a voxel divided by the standard deviation of the signal outside of the brain mask, then parcellated to each of the 88 regions of interest. We correlate fMRI SNR with all time-series features used in the analyses, and find that 2 454 time-series features are significantly correlated with SNR ($p < 0.05$, Bonferroni-corrected). We therefore regress SNR from these 2 454 time-series features and repeat the analysis in Fig 2D (S14A and S14B Fig). We find that the dynamical phenotypes of both long-lifetime PSD95 and SAP102 synapes are replicable after removing the effects of fMRI SNR. More specifically, the dynamical phenotype of long-lifetime PSD95 synapses is still characterized by measures of stationarity, including StatAv10 (S14C Fig), and the dynamical phenotype of SAP102 synapses is still characterized by measures of the timing of extreme events, including DN_OutlierInclude_n_001_mrmd (S14D Fig).

Third, to ensure that the relationship between synapse density and dynamics is not driven by motion, we correlate every regional time-series with frame-wise displacement (FD; sum of absolute translational and rotational displacement between consecutive points in time; S15 Fig). A large correlation would indicate that signal amplitude reflects motion. Across all ten mice and 88 brain regions per mouse, the range of correlations is (–0.17,0.24), and the median is 0.03. Therefore, we do not find evidence that signal amplitude is driven by motion. Furthermore, regions with high long-lifetime PSD95 synapse density and more non-stationary signal are regions with smaller correlations between motion and FD (correlation between long-lifetime PSD95 synapse density and average correlation between signal amplitude and FD: $r = -0.45$). Likewise, regions with high SAP102 synapse density and distinctive peaks in signal amplitude do not demonstrate greater correlations with FD ($r = 0.09$).

Fourth, we ensure the relationship between synapse density and dynamics is invariant to global signal regression. We apply global signal regression (GSR) to the fMRI time-series of each mouse and recompute the complete time-series phenotype (i.e. Fig 2C). We find that the time-series phenotype from global signal regressed and non-global signal regressed time-series are highly correlated ($r = 0.75$, $p \approx 0$; S16A Fig). Next, we re-correlate each time-series feature with all five synapse types. Overall, we find that synapse-related time-series profiles are highly consistent ($0.58 < r < 0.90$, S16B Fig). These analyses demonstrate that time-series phenotypes are robust to GSR.

Fifth, we assess the reproducibility of time-series features by comparing individual-mouse phenotypes to group-average phenotypes. For every brain region and mouse, we correlate the full regional time-series phenotype profile of the mouse with the average regional time-series phenotype profile ($6 471 \times 1$ vector of time-series feature values) (S17 Fig). All individual time-series phenotypes are positively correlated with the mean time-series phenotype (range of mean correlation: [0.33,0.68]). Next, for every time-series feature and mouse, we correlate the spatial distribution of time-series feature with the average time-series feature. The empirical mean correlation between individual and group time-series spatial distributions is 0.43 (see *Methods* for more details).

PLOS Biology

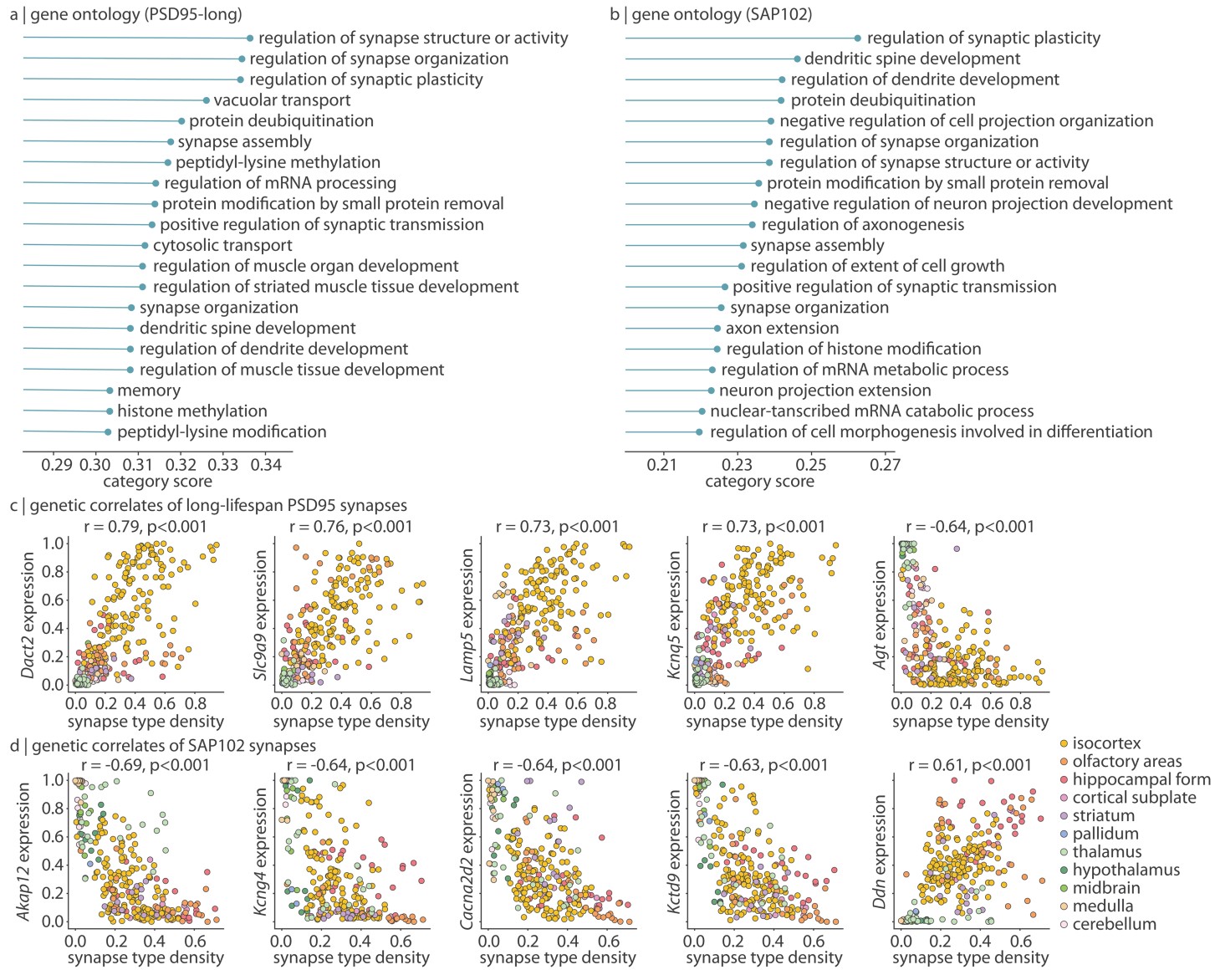

**Fig 5. Transcriptomic profiling of synapse types.** Gene expression data was acquired from the Allen Mouse Brain Atlas and correlated with synapse type density. (a–b) For each of 1 616 biological process categories associated with at least 100 genes, we calculate the median absolute correlation (Spearman's *r*) between long-lifetime PSD95 (a) or SAP102 (b) synapse density and all genes in the category ("category score", *x*-axis). The top 20 category scores for each synapse type are shown. (c–d) Selected genes whose expression is highly correlated (Spearman's *r*) with long-lifetime PSD95 (c) or SAP102 (d) synapse density. Each point is a brain region ($N = 275$) and are coloured according to major ontological structure.

Fourth, we test the specificity of the relationship between synapse type and time-series features. In this report, we find evidence that long-lifetime PSD95 synapses are associated with signal stationarity, and SAP102 synapses are associated with the timing and presence of signal outliers. But are synapse types simply coexpressed with specific cellular populations that themselves induce these dynamics? To address this question, we correlate all three synapse types with the density of 9 different cell types (neurons, glia, excitatory neurons, inhibitory neurons, modulatory neurons, astrocytes, oligodendrocytes, microglia, and all cells; S18 Fig). Long-lifetime PSD95 synapses are significantly correlated with microglia

density ($r = 0.49$, $p = 0.008$) and SAP102 synapses are significantly correlated with neurons ($r = 0.46$, $p = 0.012$) and excitatory neurons ($r = 0.60$, $p = 0.0001$). However, microglia density is only correlated with seven time-series features related to signal scaling, neuron density is not significantly correlated with any time-series features, and excitatory neuron density is only significantly correlated with a single time-series feature related to signal entropy. We therefore conclude that the reported associations between synapse type and regional dynamics are not driven by the spatial localization of specific cells.

## Discussion

In the present study, we find that different properties of neural activity can be traced back to the expression of specific synapse types. As a result, the synaptic composition of a brain region is not only associated with its macroscale dynamics but also its embedding in whole-brain structural and functional networks. Synapse lifetime emerges as an important property of the synapse, whereby long-lifetime synapses tend to be related to brain connectivity while short-lifetime synapses are involved in functional processing during wakefulness. Altogether, this work bridges the microscale and the macroscale in describing synaptic influence on whole-brain dynamics and connectivity.

The present report is part of a broader effort to understand how different synapses shape brain function. Although synapses are traditionally thought to belong to one of three groups (excitatory, inhibitory, modulatory), mass spectrometry and immunoblotting experiments have revealed an enormous diversity of synaptic proteins and synaptic functions [1,5,32]. Here we focus on two types of excitatory synapses: those that express PSD95 and those that express SAP102. These two proteins are differentially expressed throughout the brain and have different effects on the synapse [1]. While the structure and function of individual synapse types have been described [8,33,34], their macroscale impact is less well understood [1,10]. The fundamental question that we address is: is the synaptic make-up of a brain region related to its functional dynamics?

There is a rich body of literature aimed at understanding the molecular and cellular origins of neural dynamics [35–37]. Previous studies have reported that neural oscillations may emerge from specific cell types [38], local gene transcription [39], or neurotransmitter receptors [40]. However, these findings are based only on a small number of time-series features. To address this limitation, new feature engineering methods have been developed to make it possible to comprehensively summarize the statistical features of an activity time-course [19,20]. Using such an approach, recent work has demonstrated that different properties of neural dynamics can be traced back to specific elements of brain organization, including anatomical wiring [16], cell type distributions [41], and myelination [42], although how these layers of brain organization modulate dynamics is still unclear.

One overlooked yet compelling molecular feature that likely modulates regional activity is the synapse [36]. Here we compare regional dynamical phenotypes to the underlying expression of different synapses and find that each synapse type is associated with specific features of brain activity. Pharmacological experiments at the level of the single synapse have established that synapse types are associated with different excitatory postsynaptic potentials (EPSPs) [6,8]. It therefore follows that patterns in macroscale dynamics emerge as a sum of synapse-specific EPSPs. Indeed, we find that PSD95 synapses are enriched in regions with non-stationary dynamics and SAP102 synapses are enriched in regions whose dynamics are marked by high-amplitude events. This suggests that SAP102 synapses are tuned to co-incident input stimuli that result in a burst of activity, whereas PSD95 synapses are responsive to a wide range of inputs, resulting in more variable excitatory postsynaptic potentials, firing rate, and macroscale neural dynamics. Our finding is consistent with reports from single-synapse pharmacological experiments that show that PSD95 synapses are involved in synapse strengthening regardless of the input stimulus frequency [6], whereas SAP102 synapses are frequency-specific in their modulation of plasticity [8]. Altogether, this supports the notion that the overlapping distributions of heterogeneous synapse types, each associated with specific neuronal response properties, may combine to encode diverse types of inter-regional signaling and generate unique spatiotemporal dynamics [1].

The present report relates local synaptic morphology to statistical properties of regional haemodynamics, but does not experimentally address intermediate steps. Given the reported findings are derived from fMRI, which is influenced not only by neuronal activity but also by vasculature [43,44], elucidating the neuronal component of the synapse-dynamics relationship remains an important goal. This work would therefore benefit from multi-scale experiments that seek to understand how combinations of synapses generate unique neuronal population dynamics. For example, extending single-synapse pharmacological experiments to neuronal populations with a controlled synaptic composition would aid in establishing how individual EPSPs are summed over a neuronal population to generate population dynamics. Likewise, replicating this study with alternative, more direct measurements of neural activity (e.g. calcium imaging in mice, or magneto/electro-encephalography (M/EEG) in humans) would establish whether the reported time-series feature associations can be generalized to other imaging and recording technologies each with their own biological interpretation. This work would also be complemented by a deeper understanding of the relationship between synapse density and variations in regional blood flow, or more generally on neurovascular coupling, for the purposes of establishing neuronal specificity of synapse-dynamics relationships. For example, a replication analysis using time-varying measurements of cerebral blood flow would reveal whether synapse density is uniquely correlated with hemodynamic signal or if synapse density is similarly associated with variability in regional blood flow. While the relationship between neuronal activity and vascular variability is not the focus of this manuscript, we note that interpreting synapse-hemodynamic relationships is incomplete without considering neurovascular coupling [45–47]. We hope that our findings will initiate deeper exploration into the relationship between regional synaptic composition and population dynamics.

If synaptic architecture modulates regional dynamics, how then does synaptic architecture influence a brain region's embedding and participation in global structural and functional networks? We find that SAP102 and PSD95 synapses make opposite contributions to global network topology. SAP102 synapses are present in regions that make many efferent structural connections, while PSD95 synapse density is correlated with functional hubs. Furthermore, the placement of structural and functional hubs specifically reflects the presence of synapses that stably express PSD95 over the course of multiple days, weeks, or months [2]. These findings contribute to a growing body of literature that highlights the promise of enriching macroscale brain networks with microscale molecular information in revealing new insights about brain organization [25]. For example, previous work has reported that whole-brain network organization can be mapped to patterns of gene expression [48], protein abundance [40], cell type distributions [49,50], and different types of oscillatory dynamics [51,52]. Synapse architecture fills in the spaces between these other layers of description, thereby contributing to a more complete understanding of multiscale brain organization and the relationship between brain structure and function.

Why might some synapses express proteins with long lifetime while others express proteins with short lifetime? In general, protein turnover is an important process necessary for maintaining cellular health and tuning protein levels to a specific context [53,54]. Indeed, protein turnover in synapses controls synaptic plasticity and may be a molecular marker of memory, such that synapses expressing long-lifetime proteins "hold on" to adaptations whereas synapses with short-lifetime proteins quickly reset [2,55]. PSD95 lifetime in particular has been shown to be greatest in superficial layers of the isocortex (where longterm memories are stored [56]) and increases with age, presumably reflecting increased storage of long-term memory in late adulthood [57]. Meanwhile, short-lifetime PSD95 synapses are most expressed in the medulla, and during early development. The localization of long-lifetime PSD95 synapses in the isocortex and short-lifetime PSD95 synapses in the brainstem has led researchers to speculate that long-lifetime synaptic proteins are more involved in higher-order cognitive functions while short-lifetime synapses are related to innate behaviours [2]. However, we find that long-lifetime PSD95 synapses appear to play a similar role regardless of whether the mouse is awake or anaesthetized, whereas short-lifetime PSD95 synapses improve structure-function coupling specifically when mice are awake (Fig 4). We therefore hypothesize that short-lifetime PSD95 synapses are involved in the moment-to-moment cognitive flexibility required during wakefulness, whereas long-lifetime PSD95 and SAP102 synapses support more fundamental functions related to homeostasis.

Synapse diversity, arising from the many possible combinations of postsynaptic proteins, creates a large repertoire for synaptically-coded physiological responses [1,58]. It is therefore puzzling that, rather than synergistically creating a new synapse type with a unique physiological and haemodynamic profile, PSD95-SAP102 colocalized synapses appear to be redundant copies of either PSD95-exclusive or SAP102-exclusive synapses. Previous work has shown that the molecular composition of PSD95 synapses are reprogrammed in mice that lack SAP102 synapses, perhaps in a compensatory action [1,58,59]. Colocalized synapses may therefore provide the molecular "option" to reprogram a synapse from PSD95 to SAP102, or vice versa. On the one hand, synaptic reprogramming may only be necessary in the extreme case of genetic mutation that renders a synapse type damaged [59]. However, it is also possible that colocalized synapses are able to dynamically adjust their physiological response profile, according to external input. Overall, studying synaptome reprogramming in colocalized synapses will help create a better understanding of the purpose and function of colocalized synapses.

One promising future research direction in combining nanoscale synaptome mapping with macroscale whole-brain imaging is that of studying disease pathology. Synaptic pathology is a major cause of psychiatric, neurological, and developmental disorders affecting individuals across all ages [60]. Hundreds of genetic disorders target specific synaptic proteins and therefore specific synapse types [59,60]. For example, both PSD95 and SAP102 synapses are targets of genetic disorders including childhood learning disabilities [61–64]. By mapping synapse types to brain dynamics and connectivity, we will develop a deeper understanding of how pathology in specific synapse types manifests throughout the brain. Ultimately, future studies combining synaptome mapping with brain imaging, in humans and model organisms, may enhance our understanding of how imaging can be used to monitor disease progression and therapeutic interventions.

We close with some methodological limitations. First, we only consider two excitatory synapse types. While PSD95 and SAP102 are both key synaptic proteins, the diversity of synapse types is only beginning to be fully appreciated and understood [1,9,65]. Extensive research is necessary to comprehensively characterize the spatial distribution, structure, and function of the many types of synapses that exist in the brain. Second, although whole-brain synapse mapping technology is both recent and state-of-the-art, current whole-brain synapse density maps are limited to a single male mouse. This is an important limitation that hinders our ability to assess whether the current findings can be generalized to a broader population of animals or across sexes. As synapse mapping technology becomes more high throughput, it will become possible to study how synaptic architecture influences individual-specific structure, function, and behaviour. Similarly, time-series feature calculation was conducted on fMRI data from a small sample of 10 mice, and individual reproducibility of the spatial distribution of time-series features is poor (mean r=0.13). We therefore average time-series features across mice to maximize signal and reproducibility (predicted mean r=0.42; see Methods for details). In addition, fMRI data were acquired in exclusively male mice, precluding an understanding of how these findings generalize to female mice. Third, synapse density, tract-tracing and fMRI data were derived in separate mouse populations, highlighting the need for more comprehensive datasets that include measurements from diverse scales of brain organization. Furthermore, integrating measurements across spatial scales necessitated downsampling the high resolution synapse density data. Fourth, the present report uses fMRI dynamics as a proxy for neural dynamics, despite known vascular contributions to the fMRI signal [43,44] The reported findings are therefore not reflective of purely neuronal activity.

In summary, we demonstrate that local synapse morphology is robustly associated with properties of regional fMRI dynamics. As a result, the synaptic composition of a brain region will affect its participation in whole-brain structural and functional networks, as well as in different behavioural states. Altogether, this work illustrates the fundamental role of synapses in shaping whole-brain organization.

## Methods

### Synapse mapping

Synapse density data was originally acquired and shared by [1]. Briefly, the authors developed a novel synapse mapping pipeline ("SYNMAP") to image and quantify multiple synaptic parameters for two synapse types: synapses that express the synaptic protein PSD95 (Postsynaptic Density 95) and those that express SAP102 (Synapse-Associated Protein 102). Both PSD95 and SAP102 are proteins postsynaptically expressed on excitatory neurons. They are members of the membrane-associated guanylate kinase (MAGUK) superfamily and function as scaffolding proteins which assemble neurotransmitter receptors, ion channels, and other structural and signaling proteins into a multiprotein signaling complex.

The authors use a single adult (postnatal day 80 male) genetically engineered mouse for whom its PSD95- and SAP102-coding genes (*Dlg4* and *Dlg3*) have been tagged with fluorescent proteins (eGFP and mKO2, respectively) such that the synaptic proteins are expressed alongside fluorescent proteins. All mouse procedures were performed in accordance with UK Home Office regulations and approved by Edinburgh University Director of Biological Services. The whole brain was dissected out and five 18 μm thick coronal slices were sectioned using a NX70 Termo Fisher cryostat. Fluorescently labelled proteins were imaged using a Spinning Disk confocal Microscopy (SDM) platform and quantified using an image detection algorithm, Ensemble Detection [1,66,67]. From here, synapses were classified as either expressing only PSD95, only SAP102, or both PSD95 and SAP102.

Next, six punctum parameters were quantified for each detected and localized synaptic puncta: mean punctum pixel intensity, punctum size, skewness, kurtosis, circularity, and aspect ratio (the latter four being measurements of punctum shape). Synaptic puncta parameters were averaged within each volumetric area (19.2 μm × 19.2 μm × 0.5 μm), then averaged over all volumetric areas within each of 775 anatomical regions defined by the Allen Reference Atlas (ARA) [68]. A weighted ensemble clustering (WEC) algorithm was applied to further cluster PSD95 synapses into 11 subtypes, SAP102 synpases into 7 subtypes, and PSD95/SAP102 colocalized synapses into 19 subtypes, according to their synapse morphology. For each subtype, their density (number of detected and localized synaptic puncta per unit area) was mapped across the 775 regions of the ARA, resulting in a 37 × 775 matrix of synapse densities (shown in Fig 1A). We confirm synapse density is highly symmetric (S19 Fig).

Rather than analyzing each subtype separately, we group subtypes according to how similarly they are anatomically expressed (Fig 1B). We apply a consensus Louvain clustering algorithm ($\gamma = 1$, 250 repetitions [69,70]) to the synapse similarity matrix, stratified by synaptic protein (i.e. separately clustered for all PSD95-expressing synapses, SAP102-expressing synapses, and colocalized synapses) and find two PSD95 clusters, one SAP102 cluster, and three PSD95/SAP102 colocalized clusters, one of which is made up of a single synapse subtype which we omit from future analyses. Since SAP102 subtypes are all highly coexpressed, we average them together into a single mean SAP102 synapse density map (shown in Fig 1G). PSD95 subtypes follow one of two expression patterns, which nearly perfectly divide PSD95 subtypes according to their PSD95 lifetime (save for one PSD95 subtype which has a short lifetime but is included in the "long-lifetime" cluster; Fig 1B barplot). We therefore average PSD95 subtypes within each cluster into a long-lifetime PSD95 synapse density map (Fig 1E) and a short-lifetime PSD95 synapse density map (Fig 1F). Note that throughout, the labels "long-lifetime PSD95 synapses" and "short-lifetime PSD95 synapses" refer to the lifetime of the protein (PSD95) and not the lifetime of the synaptic connection. Colocalized synapse subtypes are averaged within each cluster. We find that one cluster ("SAP102-like colocalized synapses") is highly coexpressed with SAP102 synapses, and the other ("PSD95-like colocalized synapses") is highly coexpressed with long-lifetime PSD95 synapses (S1 Fig). Finally, for the analyses in the present report, the high spatial resolution synapse density data had to be downsampled. When applicable, synapse density was averaged across anatomical regions that fall within a single parent region, resulting in 88 bilateral regions when comparing with functional data, 137 right hemisphere regions when comparing with structural data, and 275 right hemisphere regions when comparing with gene expression data. For layer-specific analyses in Fig 3C, 31 isocortical right hemisphere regions were available in layers I, V, VI, 30 were available in layer II/III, and 15

 

were available in layer IV. All mouse brains were plotted using `brainglobe-heatmap` (https://github.com/brainglobe/brainglobe-heatmap) [3].

**Synapse protein lifetime quantification**

Synapse protein lifetime data was collected by the same group that collected the synapse density data, as reported in [2]. Briefly, the authors used HaloTag technology to measure PSD95 lifetime in PSD95-expressing synapses throughout the brain. HaloTag involves tagging PSD95 with a HaloTag domain, which then forms an irreversible bond with a HaloTag ligand, when the ligand is injected into the brain. By coupling the HaloTag ligand with a fluorophore, the authors are able to time the labelling of PSD95 synapses (according to when the injection of the HaloTag ligand occurs), then wait a defined amount of time before sacrificing the animal and imaging the synaptic puncta, using the methodology described in *Synapse mapping*. The number of synaptic puncta identified at time $T$ is compared to the number of synaptic puncta identified immediately after injection, resulting in a measurement of the percentage of PSD95 synapses remaining after time $T$. In the present study, we use synapse protein lifetime data from 6 month-old female mice ($n = 8$ mice sacrificed day 0 after injection, and $n = 8$ mice sacrificed day 7 after injection). Synapse protein lifetime is defined as the percentage of PSD95 synapses remaining in the whole-brain at day 7 compared with day 0 (Fig 4A in [2]), and is quantified for each of the 11 PSD95 synapse subtypes.

**fMRI data acquisition**

Resting-state fMRI scans were acquired by [17] in 10 awake, head-fixed C57BL/6J male mice (mean $\pm$ standard deviation of age = $141.4 \pm 21.2$ days; full methodological details available in [17]). First, a surgery was conducted to attach a custom-made headpost to the skull. This headpost was designed such that the mice could be secured to a custom-made MRI-compatible animal cradle which ensured immobilization during scanning. 10–15 days after the headpost surgery, the mice followed a habituation protocol to become acclimatized to the scanning procedure. The habituation protocol involved multiple days of habituation to the experimenter, the cradle, head fixation, scanning environment, and scanning sounds. All fMRI scans were acquired on a 7.0 Tesla MRI scanner (Bruker Biospin, Ettlingen) with a BGA-9 gradient set, a 72 mm birdcage transmit coil, and a four-channel solenoid receive coil. A single-shot echo planar imaging (EPI) sequence was acquired with the following parameters: TR/TE=1000/15ms, flip angle=60°, matrix=100 × 100, FOV=2.3×2.3 cm, 18 coronal slices (voxel-size: 230 × 230× 600 ☐m), slice thickness=600 ☐m, and 1920 time points (32 minutes total).

BOLD data preprocessing involved removing the first 2 minutes of every time-series, time despiking, motion correction, skull stripping, and spatial registration to an in-house mouse brain template with spatial resolution of $0.23 \times 0.23 \times 0.6$ mm$^3$ [71]. Denoising and motion correction involved regressing out average cerebral spinal fluid signal, as well as 24 motion parameters determined from the 3 translation and rotation parameters estimated during motion correction, their temporal derivatives, and corresponding squared regressors. No global signal regression was performed. Frame-wise displacement (FD)-derived measurements of in-scanner head motion were not significantly different as those obtained in anaesthetized animals. Furthermore, frame-wise fMRI scrubbing was employed, using an FD threshold of 0.075 mm. Finally, the time-series were band-pass filtered ($0.01 - 0.1$ Hz) and spatially smoothed with a Gaussian kernel of 0.5 mm full width at half maximum.

Two additional groups of age matched male C57BL/6J mice were scanned under anaesthesia in [17]. The first group includes 19 mice (mean $\pm$ standard deviation age = $77.4 \pm 6.9$ days) scanned under shallow halothane anesthesia (0.75%) [31]. The second group includes 14 mice (mean $\pm$ standard deviation age = $168 \pm 57.4$ days) scanned under medetomidine-isofluorane anesthesia (0.05 mg/kg bolus and 0.1 mg/kg/h IV infusion, plus 0.5% isofluorane). All imaged mice, awake and anesthetized, were imaged in the same lab, bred in the same vivarium, and scanned with the same MRI protocol.

PLOS Biology

Finally, fMRI volumes for each mouse was parcellated to 162 regions defined by the ARA, of which only 88 regions were also present in the synapse density data. Due to motion scrubbing, different mice had a different number of time-points, with the minimum being 1414; therefore, time-series for the remaining mice were all truncated from the end to 1414 time-points. The final fMRI data for all mice had size 88 regions × 1414 time-points.

**Time-series feature extraction**

For every mouse and every brain region, the corresponding time-series was subjected to an automated massive time-series feature extraction pipeline using the Highly Comparative Time-Series Analysis toolbox (*hctsa*, https://github.com/benfulcher/hctsa [19,20]). This toolbox calculates > 7 000 statistical features (e.g. mean, variance, stationarity, entropy) on each time-series and normalizes the features across brain regions to the unit interval using a scaled robust sigmoid normalization (parameter `normFunction`= 'mixedSigmoid'). After discarding features with zero variance across regions, we retained the subset of features present across all mice, resulting in a 88 region × 6471 feature × 10 mice matrix of time-series feature values.

To assess the reproducibility of the spatial distribution of each time-series feature, we calculate pairwise spatial correlations for every time-series feature across every pair of mice. We find that time-series reproducibility is low (mean Spearman's $r = 0.13$). We therefore average time-series feature values across mice to maximize signal and reproducibility. Indeed, when we correlate the spatial distributions of each single-mouse time-series feature with the group-averaged spatial distribution, we find increased reproducibility (mean Spearman's $r = 0.43$). We take this one step further and assess the reproducibility of group-mean spatial distributions of time-series features as a function of sample size. More specifically, from our sample of $N = 10$ mice, we calculate the average spatial distribution of all temporal features across every distinct combination of $N = 2, 3, 4, 5$ mice. For every pair of groups, we correlate their group-mean time-series feature spatial distribution, and finally calculate the average across all pairwise correlations (i.e. mean reproducibility). We find that mean reproducibility increases with sample size logarithmically ($R^2 = 0.9998$) and predict that the mean reproducibility when $N = 10$ is 0.42 (see S20 Fig). We therefore only analyze group-mean time-series features, resulting in a 88 region × 6 471 feature matrix of time-series feature values (shown in Fig 2C).

We next correlate each time-series feature with synapse density. After Bonferroni multiple comparisons correction, we find 26 features are significantly correlated with long-lifetime PSD95 synapse density, 6 features are significantly correlated with short-lifetime synapse density, and 211 features are significantly correlated with SAP102 synaspe density (see Supplementary Table S1 for feature names, correlation coefficient, and corrected *p*-value). Due to the small number of features that are correlated with short-lifetime PSD95 synapses, as well as their relatively small correlation coefficient (all with magnitude <0.5) and large *p*-value (none <0.01), we do not further analyze features associated with short-lifetime PSD95 synapses.

For the other two synapse types, we calculate a feature × feature similarity matrix for significantly correlated features (in the case of SAP102, we also threshold these features to those where *r* > 0.5, to reduce the set from 211 features to 59). This similarity matrix was clustered nine times using agglomerative clustering with the number of clusters defined as 2 through 10. Clustering solutions were visually inspected and one solution is shown in S6A Fig (long-lifetime PSD95) and S5A Fig (SAP102). This clustering solution was selected based on (1) minimizing the number of clusters with very few (<5) features, (2) maximizing the correlation within clusters, and (3) minimizing the correlation between clusters. Finally, we triangulate toward a single representative time-series feature to highlight in the main text by considering (1) correlation coefficient magnitude, (2) number of features that measure a similar property, which can be estimated via cluster size, and (3) feature explainability.

This similarity matrix was clustered into 2–10 clusters using agglomerative clustering and we show one clustering solution in S3A Fig (long-lifetime PSD95) and S4A Fig (SAP102). We triangulate toward a single representative feature to

highlight in the main text by considering (1) correlation coefficient magnitude, (2) number of features that measure a similar property, which can be estimated via cluster size, and (3) feature explainability.

For long-lifetime PSD95 synapses, we selected `StatAv10` [21], which is calculated by binning the time-series into ten non-overlapping segments, calculating the mean of each segment, calculating the standard deviation across the ten segment means, and normalizing this standard deviation with the standard deviation of the full time-series (S3B Fig). For SAP102 synapses, we selected `DN_OutlierInclude_n_001_mrmd` which, for all "extreme" events below a moving threshold (in this case, thresholds are all $\leq 0$), calculates the mean time at which all below-threshold events occur (time being normalized such that -1 is the start of the time-series, 0 is the middle, and 1 is the final time point), then calculates the median mean time across all thresholds (S4B Fig).

## Structural data acquisition

The Allen Mouse Brain Connectivity Atlas was first presented in [22] and maps axonal projections for 295 ARA-defined brain regions in C57BL/6J male mice at postnatal age P56. Each mouse brain was sectioned into 140 slices (0.35 μm $x$–$y$ resolution, 100 μm thickness), and a eGFP-labelled anterograde tracer was injected into multiple anatomical regions on each slice. Each slice was registered to the Allen Reference Atlas at a resolution of 100 μm$^3$. In total, 469 viral microinjection experiments were conducted to map axonal connectivity.

Next, [22] summarized the connectivity data in a weighted, directed adjacency matrix with includes a normalized connection strength and an associated $p$-value for ipsilateral connections in the right hemisphere and contralateral connections from right to left hemisphere (Supplementary Table 2 of [22]). 80 brain regions were excluded due to insufficiently labelled voxels from the injection experiment, and an additional 2 regions were excluded because they were not linearly separable under the connectivity model given the full dataset, resulting in a final connectivity matrix representing 213 source regions, 213 ipsilateral target regions, and 213 contralateral target regions.

In the present report, we focus solely on ipsilateral connectivity in the right hemisphere ($213 \times 213$ weighted directed adjacency matrix). As per [72], we omit connections where $p < 0.05$ (excluding self-connections), resulting in a connection density of 6.9%. Finally, we reduced the set of regions to those also present in the synapse density data, resulting in a $137 \times 137$ weighted and directed structural connectivity matrix.

## Structure-function coupling

Structure-function coupling at every brain region is defined as the adjusted $R^2$ of a simple linear regression model that fits regional communicability (i.e. the communicability between a brain region to every other brain region) to regional functional connectivity (i.e. the functional connectivity between a brain region and every other brain region). Communicability is defined as the weighted average of all walks and paths between two brain regions, as defined on the structural connectivity matrix, and represents diffusive communication [73,74]. Additionally, communicability has been previously demonstrated as an important bridge between brain structure and function [26,27,40,75]. In the synapse-informed model, regional synapse type density for long-lifetime PSD95, short-lifetime PSD95, and SAP102 synapses separately, is included as an independent variable. The change in fit ($\Delta R^2_{adj}$) is defined as the difference between $R^2_{adj}$ before and after adding synapse type density to the model. The distribution of $\Delta R^2_{adj}$ for all regions was compared across states (awake versus anaesthetized) using a two-sided dependent non-parametric t-test for paired samples (Wilcoxon signed-rank test). To conduct this analysis, we used the subset of brain regions present in the structural, functional, and synapse density datasets, resulting in 35 right hemisphere brain regions.

## Gene expression data acquisition

In situ hybridization data were obtained from the Allen Mouse Brain Atlas for adult C57BL/6J male mice (age P56) [23]. Using the `abagen` toolbox (https://abagen.readthedocs.io/en/stable/ [76]), gene expression density data (proportion of

expressed voxels in an anatomical division) was downloaded for the 275 unique Allen Reference Atlas regions for which synapse density data are also defined. Data were averaged across experiments, and were separately downloaded for data from sagittal (19 919 genes) and coronal (4 083 genes) brain sections. Gene expression was normalized using a robust sigmoid transformation [19,72], such that

$$x_{norm} = \frac{1}{1 + \exp(-\frac{(x - \langle x \rangle)}{IQR_x})}$$

where $x_{norm}$ is the normalized expression value of the gene, $\langle x \rangle$ is the median and $IQR_x$ is the normalized interquartile range of gene expression across regional samples. Normalized expression values were then rescaled to the unit interval:

$$x_{scaled} = \frac{x_{norm} - \min(x_{norm})}{\max(x_{norm}) - \min(x_{norm})}$$

For univariate correlations between gene expression and synapse density, we only consider genes whose expression is (1) measured in both sagittal and coronal sections, and (2) the correlation of gene expression from sagittal and coronal sections is $\geq 0.70$, resulting in a total of 1 295 genes. The correlational analysis uses gene expression measurements from coronal sections. For the gene ontology analysis in Fig 5C and 5D, we use the full set of genes acquired using sagittal sections.

## Gene ontology analysis

To understand the function of genes that are coexpressed with specific synapse types, we conduct a gene category enrichment analysis. We use Gene Ontology biological processes and annotations, downloaded from https://zenodo.org/records/4460714 [77] (see also https://github.com/benfulcher/GeneCategoryEnrichmentAnalysis [78]), which were originally downloaded directly from https://geneontology.org/docs/go-enrichment-analysis/ [79]. This database includes 30 248 biological processes, each associated with a list of genes. We calculate a "category score" for each biological process, defined as the median absolute Spearman's correlation between a gene's expression and synapse density, across all genes in the category. We only consider categories with at least 100 associated genes, resulting in 1 616 categories.

## Cell type density

Density (mm$^3$) of cells (all), neurons, glia, excitatory cells, inhibitory cells, modulatory cells, astrocytes, oligodendrocytes, and microglia in the mouse brain are derived from Nissl microscopy data in the Allen Mouse Brain Atlas [23] and can be found in supplementary data sheet 2 of [80]. Since region names in the cell density dataset do not include hemisphere, we compare the brain regions in the cell density atlas with the 44 name-matched left hemisphere regions from the functional parcellation (Fig 1D). Correlations are shown in S18 Fig.

## Supporting information

**S1 Table. Spearman's correlation and Bonferroni-corrected p-value for statistically-significant correlations between time-series features and synapse type density.**
(XLSX)

**S1 Fig. Colocalized synapse type density in the mouse brain.** We extend Fig 1 to show brain maps of colocalized synapse types. (a–b) Axial, coronal, and sagittal view of mean density of the first cluster (a) and second cluster (b) of PSD95/SAP102 colocalized synapses, mapped to both 137- and 88-region parcellations. The colourbar in (b) applies to panels a–b. (c–e) Scatter plot showing the correlation between long-lifetime PSD95-exclusively expressing synapses and

PSD95-like (cluster 2) colocalized synapses, as well as between SAP102-exclusively expressing synapses and SAP102-like (cluster 1) colocalized synapses. Correlations are shown using all 775 regional samples (c), the structural parcellation (d), and the functional parcellation (e). All mouse brains are plotted using `brainglobe-heatmap` [3].
(EPS)

**S2 Fig. Parcellated synapse type density in the mouse brain.** We repeat what is shown in Fig 1A and 1B, parcellated to (a) the 137-region right hemisphere parcellation and (b) the 88-region bilateral parcellation. Regions in the lefthand synapse density matrix are ordered by ontological structure, as indicated by the colours in the horizontal bar. The right-hand similarity matrix represents how similarly (Spearman's $r$) pairs of synapse subtypes are spatially expressed. Clusters are derived using the Louvain community detection algorithm on the original 775-region matrix (Fig 1A) [69,70].
(EPS)

**S3 Fig. Time-series features most associated with long-lifetime PSD95 synapse density.** We correlate long-lifetime PSD95 synapse density with each of 6 471 time-series features. (a) We generate a feature × feature correlation matrix (Spearman $r$) for the 26 time-series features that are significantly correlated with long-lifetime PSD95 synapse density after Bonferroni correction. The correlation between each time-series feature and long-lifetime PSD95 synaspe density is shown in the scatter plot on the right of the matrix. One feature (`StatAv10` [21]) is selected as a representative feature in the main text. Black boxes represent clusters identified using agglomerative clustering. (b) `StatAv10` is calculated by dividing a z-scored time-series into ten non-overlapping bins (dashed vertical lines), calculating the mean signal amplitude within each bin, normalized by the standard deviation of the full time-series (green points), then calculating the standard deviation across normalized bin means. We illustrate this process for two example time-series, one (endopiriform nucleus) with low long-lifetime PSD95 synapse density (and low `StatAv10`), and the other (primary somatosensory area, barrel field) with large long-lifetime PSD95 synapse density (and high `StatAv10`).
(EPS)

**S4 Fig. Time-series features most associated with SAP102 synapse density.** We correlate SAP102 synapse density with each of 6 471 time-series features. (a) We generate a feature × feature correlation matrix (Spearman $r$) for the 59 time-series features that are both significantly correlated with SAP102 synapse density and $r > 0.5$. The correlation between each time-series feature and SAP102 synaspe density is shown in the scatter plot on the right of the matrix. One feature (`DN_OutlierInclude_n_001_mrmd`) is selected as a representative feature in the main text. Black boxes represent clusters identified using agglomerative clustering. (b) `DN_OutlierInclude_n_001_mrmd` is calculated by defining a threshold from 0 to -3.5 (0.01 increments; $x$-axis of the green plot; example threshold shown as the horizontal dashed line in the time-series plots at the top and as a vertical dashed line in the green plots at the bottom), selecting all time-points where the z-scored time-series' value is below the threshold (pink points on the time-series; time is normalized from -1 to 1), calculating the mean time at which these extreme events occur ($y$-axis of the green plot), and finally calculating the median mean time across all thresholds. We illustrate this process for two example time-series, one (midbrain, motor area) with low SAP102 synapse density (and a lower `DN_OutlierInclude_n_001_mrmd`, that is, an earlier occurrence of extreme events) and one (piriform-amygdalar area) with high SAP102 synapse density (and a higher `DN_OutlierInclude_n_001_mrmd`, that is, a later occurrence of extreme events).
(EPS)

**S5 Fig. Time-series features most associated with SAP102-like colocalized synapse density.** We repeat the analysis in S4 Fig using SAP102-like colocalized synapses. We generate a feature × feature correlation matrix (Spearman $r$) for the 46 time-series features that are both significantly correlated with SAP102-like colocalized synapse density and $r > 0.5$. The correlation between each time-series feature and SAP102-like colocalized synaspe density is shown in the scatter plot on the right of the matrix. The feature with the greatest correlation is `SY_LocalGlobal_std_p1` which compares the standard deviation of the first 1% of the time-series with the standard deviation of the full time-series.

`DN_OutlierInclude` reflects a similar phenomenon except rather than measuring standard deviation `DN_OutlierInclude` considers the presence and timing of outlier events. In other words, `DN_OutlierInclude` allows us to better understand why the standard deviation in local segments of the time-series may differ from the global time-series. Black boxes represent clusters identified using agglomerative clustering.
(EPS)

**S6 Fig. Time-series features most associated with PSD95-like colocalized synapse density.** We repeat the analysis in S3 Fig using PSD95-like colocalized synapses. We generate a feature × feature correlation matrix (Spearman r) for the 41 time-series features that are significantly correlated with PSD95-like colocalized synapse density after Bonferroni correction. The correlation between each time-series feature and PSD95-like colocalized synapse density is shown in the scatter plot on the right of the matrix. The time-series feature with the highest correlation is `SC_FluctAnal_2_dfa_50_logi_ssr`, which considers how segments of the time-series fluctuate with respect to segment size, and whether the relationship between fluctuation and segment size scales. Signal stationarity also measures the variability of the signal across different windows, but `StatAv10` only considers one window size (10% of the time-series) whereas `SC_FluctAnal` tests varying window sizes. Black boxes represent clusters identified using agglomerative clustering.
(EPS)

**S7 Fig. Time-series and interregional connectivity profiling of short-lifetime PSD95 synapses.** We repeat what is shown in Figs 2 and 3 for short-lifetime PSD95 synapses. (a) Coronal view of short-lifetime PSD95 synapses. (b) Absolute correlation coefficient (Spearman's r) between all time-series features and short-lifetime PSD95 synapse density. Green points are statistically significant ($p < 0.05$ after Bonferroni correction). Top features are all related to model forecasting. (c) Scatter plots showing the correlation between short-lifetime PSD95 synapse density (x-axis) and weighted in-degree of SC, weighted out-degree of SC, and weighted degree of FC. Each point is a brain region, and points are coloured according to major ontological structure.
(EPS)

**S8 Fig. Colocalized synapse types colocalize with structural and functional hubs.** (a) Correlations between cluster 1 colocalized (SAP-102 like) synapses or cluster 2 colocalized (long-lifetime PSD95-like) synapse density and weighted in- or out-degree of structural connectivity (sum of all afferent or efferent connections to or from a brain region, respectively). Weighted degree (y-axis) is log-transformed for visualization. Each point is a brain region, and regions are coloured according to their ontological structure. (b) Synapse density within each of 5 layers in the isocortex were separately correlated with weighted in- and out-degree. Asterisks represent $p < 0.05$. (c) Correlations between cluster 1 (SAP102-like) or cluster 2 (PSD95-like) colocalized synapse density and weighted degree of functional connectivity (sum of functional connectivity between one region and all other regions). Each point is a brain region, and regions are coloured according to their ontological structure. In the left-most scatter plot, only isocortical, olfactory, and cortical subplate regions are outlined and included in the correlation calculation.
(EPS)

**S9 Fig. Awake vs anaesthetized change in structure-function coupling ($\Delta R^2_{adj}$) stratified by functional systems.** We compare the change in structure-function coupling (distribution of $\Delta R^2_{adj}$) in awake versus anaesthetized mice, for all three synapse types (originally shown in Fig 3C), stratified by two network systems. (a) $\Delta R^2_{adj}$ is stratified by functional system included in region name (e.g. "Primary motor area" is labelled "motor", "Primary somatosensory area, barrel field" is labelled "somatosensory"). (b) $\Delta R^2_{adj}$ is stratified by structural and functional connectivity networks defined by [81]. Data

underlying violinplots can be found at https://github.com/netneurolab/hansen_synaptome/blob/main/results/violinplot_underlying_data.xlsx. DMN-MID = midline DMN; DMN-PLN = posterior-lateral DMN; LCN = latero-cortical network.
(EPS)

**S10 Fig. Colocalized synapse type density improves structure-function coupling more in anaesthetized mice.** For each brain region, we predict its functional connectivity profile from either regional communicability alone (a measure of structural connectedness) or regional communicability alongside three synaptic variables: cluster 1 colocalized synapse density, cluster 2 colocalized synapse density, and either long-lifetime PSD95, short-lifetime PSD95, or SAP102 synapse density. Structure-function coupling is defined as the fit ($\Delta R^2_{adj}$) of the model. We compare the change in structure-function coupling (distribution of $\Delta R^2_{adj}$) in awake versus anaesthetized mice. Statistical significance is assessed using a two-sided dependent non-parametric t-test for paired samples (Wilcoxon signed-rank test). Violin plots estimate a kernel density on the underlying data, the white point represents the median, the thick vertical line represents the quartiles of the distribution, and the thin vertical line represents the range. Data underlying violinplots can be found at https://github.com/netneurolab/hansen_synaptome/blob/main/results/violinplot_underlying_data.xlsx.
(EPS)

**S11 Fig. Synapse type density improves structure-function coupling.** Structure-function coupling analysis in Fig 4 was repeated using a functional connectome from mice anaesthetized with medetomidine and isofluorane. (a) Structure-function coupling before (*x*-axis) and after (*y*-axis) adding synapse type density (long-lifetime PSD95, short-lifetime PSD95, and SAP102) to the model. Each point is a brain region, and points are coloured according to major ontological structure. The identity line is shown in grey. Coronal slices show the change in coupling ($\Delta R^2_{adj}$) after adding synapse density to the model. Data are mirrored across hemispheres for visualization. (b) We compare the change in structure-function coupling (distribution of $\Delta R^2_{adj}$) in awake versus anaesthetized mice, for all three synapse types separately. Statistical significance is assessed using a two-sided dependent non-parametric t-test for paired samples (Wilcoxon signed-rank test). Violin plots estimate a kernel density on the underlying data, the white point represents the median, the thick vertical line represents the quartiles of the distribution, and the thin vertical line represents the range. Data underlying violinplots can be found at https://github.com/netneurolab/hansen_synaptome/blob/main/results/violinplot_underlying_data.xlsx.
(EPS)

**S12 Fig. Transcriptomic profile of short-lifetime PSD95 synapses.** We repeat what is shown in Fig 5 for short-lifetime PSD95 synapses. (a) For each of 1 616 biological process categories associated with at least 100 genes, we calculate the median absolute correlation (Spearman's *r*) between short-lifetime PSD95 synapse density and all genes in the category ("category score", *x*-axis). The top 20 category scores for each synapse type are shown. (b) Selected genes whose expression is highly correlated (Spearman's *r*) with short-lifetime PSD95 synapses. Each point is a brain region, and points are coloured according to major ontological structure.
(EPS)

**S13 Fig. Transcriptomic profiling of colocalized synapse types.** Gene expression data was acquired from the Allen Mouse Brain Atlas and correlated with colocalized synapse type density. (a–b) For each of 1 616 biological process categories associated with at least 100 genes, we calculate the median absolute correlation (Spearman's *r*) between cluster 1 colocalized (SAP102-like) (a) or cluster 2 colocalized (long-lifetime PSD95-like) (b) synapse density and all genes in the category ("category score", *x*-axis). The top 20 category scores for each synapse type are shown.
(EPS)

**S14 Fig. Signal-to-noise ratio of fMRI signal in awake mice.** (a) Signal-to-noise ratio (SNR) is calculated as the mean signal within a region divided by the standard deviation of the signal outside of the brain. SNR is shown on a coronal,

axial, and sagittal slice. (b) SNR-regressed time-series features were recorrelated with long-lifetime PSD95 and SAP102 synapse density. Green points are statistically significant ($p < 0.05$ after multiple comparisons correction). (c)–(d) For all features that are significantly correlated with long-lifetime PSD95 synapse density (c) or SAP102 synapse density (d) (after SNR regression), we generate a feature × feature similarity matrix. The scatter plot on the right shows the correlation between each feature and synapse density. The highlighted feature represents that which was mentioned in the main text and in S3 and S4 Figs.
(EPS)

**S15 Fig. Relationship between signal amplitude and frame-wise displacement.** For each mouse, we correlate regional signal amplitude (88 regions) with frame-wise displacement (FD), that is, the amount of motion between any two consecutive time-points. Violin plots estimate a kernel density on the underlying data (88 correlation coefficients). The circle represents the median, the thick vertical line represents the quartiles of the distribution, and the thin vertical line represents the range. Data underlying violinplots can be found at https://github.com/netneurolab/hansen_synaptome/blob/main/results/violinplot_underlying_data.xlsx.
(EPS)

**S16 Fig. Temporal features are robust to global signal regression (GSR).** The analysis in Fig 2 was repeated after applying GSR to all regional time-series. (a) Time-series feature values without (x-axis) and with (y-axis) GSR are highly correlated (r=0.75). (b) Each synapse type is associated with a time-series phenotype (defined as the Spearman correlation between spatial distributions of synapse type and time-series features; Fig 2D). We correlate time-series phenotypes before and after GSR.
(EPS)

**S17 Fig. Regional time-series phenotype consistency across individual mice.** Each brain region is associated with a time-series phenotype ($6\,471 \times 1$ vector of time-series values). This heatmap represents the Spearman correlation between each region's group-averaged time-series phenotype with the time-series phenotypes from individual mice. Brain regions are ordered according to ontological structure, as indicated by the vertical bar and the legend.
(EPS)

**S18 Fig. Relationship between synapse type density and cell density.** For each synapse type (y-axis), we correlate its spatial distribution with the density of 9 different cell types (x-axis) [80]. Asterisks represent statistically significant associations (Bonferroni-corrected $p < 0.05$).
(EPS)

**S19 Fig. Synapse density is symmetric across hemispheres.** We correlate right and left hemisphere synapse density for all 37 synapse subtypes.
(EPS)

**S20 Fig. Inter-individual variability of time-series feature distributions across brain regions.** (a) Green distribution: pairwise Spearman correlation of time-series feature spatial distribution for every pair of mice (10 mice total = 45 pairs) and all time-series features ($6\,471$ features). Mean correlation (green dashed line) is 0.13. Blue distribution: Spearman correlation between group-average time-series feature spatial distribution and individual time-series feature spatial distribution, for all time-series features ($6\,471$ features) and mice (10 mice). Mean correlation (blue dashed line) is 0.43. (b) For $N \in 2, 3, 4, 5$, we select all possible combination of 2 size-$N$ subgroups of mice, average time-series feature spatial profiles within subgroups, and correlate the two subgroups. The average correlation across time-series features and subgroups ("reproducibility") is shown on the y-axis. Next, we fit a logarithmic function on the empirical mean reproducibility (model fit $R^2 = 0.9998$) and predict the mean reproducibility for up to 10 mice.
(EPS)

## Acknowledgments

We thank Vincent Bazinet, Eric Ceballos, Asa Farahani, Zhen-Qi Liu, Filip Milisav, Moohebat Pourmajidian, Tahmineh Taheri, and Yigu Zhou for their comments and suggestions on the manuscript.

## Author contributions

**Conceptualization:** Justine Y. Hansen, Bratislav Misic.

**Data curation:** Zhen Qiu, Silvia Gini, Alessandro Gozzi, Seth G. N. Grant.

**Formal analysis:** Justine Y. Hansen, Ben D. Fulcher.

**Investigation:** Justine Y. Hansen.

**Methodology:** Justine Y. Hansen, Andrea I. Luppi, Zhen Qiu, Silvia Gini, Ben D. Fulcher, Alessandro Gozzi, Seth G. N. Grant.

**Project administration:** Bratislav Misic.

**Resources:** Zhen Qiu, Seth G. N. Grant.

**Visualization:** Justine Y. Hansen.

**Writing – original draft:** Justine Y. Hansen, Bratislav Misic.

**Writing – review & editing:** Justine Y. Hansen, Andrea I. Luppi, Zhen Qiu, Silvia Gini, Ben D. Fulcher, Alessandro Gozzi, Seth G. N. Grant, Bratislav Misic.

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
