## [Editor Report · Decision Letter 0]

23 Jun 2025

Dear Dr Misic,

Thank you for submitting your manuscript entitled "Synaptome architecture shapes regional dynamics in the mouse brain" for consideration as a Research Article by PLOS Biology.

Your manuscript has now been evaluated by the PLOS Biology editorial staff, as well as by an academic editor with relevant expertise, and I am writing to let you know that we would like to send your submission out for external peer review.

Once your full submission is complete, your paper will undergo a series of checks in preparation for peer review. After your manuscript has passed the checks it will be sent out for review. To provide the metadata for your submission, please Login to Editorial Manager (https://www.editorialmanager.com/pbiology) within two working days, i.e. by Jun 25 2025 11:59PM.

Kind regards,

Taylor

Taylor Hart, PhD,

Associate Editor

PLOS Biology

thart@plos.org

---

## [Decision Letter · Decision Letter 1]

11 Sep 2025

Dear Dr Misic,

Thank you for your patience while your manuscript "Synaptome architecture shapes regional dynamics in the mouse brain" was peer-reviewed at PLOS Biology. It has now been evaluated by the PLOS Biology editors, an Academic Editor with relevant expertise, and by several independent reviewers.

In light of the reviews, which you will find at the end of this email, we would like to invite you to revise the work to thoroughly address the reviewers' reports.

The reviewers wrote that the study attempts to address an important and timely topic. However, they pointed out a number of limitations in the approach that raise concerns about the strength of evidence for the major claims. Although we understand that fully addressing all of the reviewers' concerns may not be feasible within the scope of a revision, you should carefully consider the comments and suggested remedies from the reviewers and respond to them thoroughly. In addition, your revised manuscript should clarify the intended scope as a research article, rather than as a resource.

Given the extent of revision needed, we cannot make a decision about publication until we have seen the revised manuscript and your response to the reviewers' comments. Your revised manuscript is likely to be sent for further evaluation by all or a subset of the reviewers.

**IMPORTANT - SUBMITTING YOUR REVISION**

*Re-submission Checklist*

*Published Peer Review*

*PLOS Data Policy*

*Blot and Gel Data Policy*

Sincerely,

Taylor

Taylor Hart, PhD,

Associate Editor

PLOS Biology

thart@plos.org

REVIEWS:

Reviewer #1: In this article, the authors aim to make an important link between microscale synaptic events and their functional embedding in global changes of brain dynamics, by leveraging advanced and cutting-edge techniques and conducting a thorough investigation and validation across multiple fMRI datasets. The main findings revolve around how specific synapse types are associated with specific features in (f)MRI dynamics and structure. This article is well written and clear in its logic and findings. I particularly appreciate the direct questions guiding each section and reminding the reader of the main goal of each subpart. I also appreciate seeking validation in other datasets, and across brain states. A few comments may help frame the results.

1) Overall, I am not fully familiar with the hctsa tool for timeseries features extraction. Yet, I feel the article often conflates correlation with causation. For example, stating that certain synapse types "shape" neural dynamics:

a. suggests a causal relationship, which cannot be firmly concluded from correlative analyses. Even more so since—as far as I understand it—this data is correlating with task-free resting state data, so it's looking at spontaneous, not evoked, fluctuations.

b. defines 'BOLD-fMRI' fluctuations as 'neural signal' which isn't entirely biologically true. The BOLD signal of fMRI is not a direct measure of underlying neuronal changes, and even when taking into account a hemodynamic response, this may change its shape across different brain regions/brain states.

The language should be more cautious and recognize the correlative nature of the findings, or if there is a de facto causal relationship, this should be stated more clearly for audience not familiar with the hctsa tool and the features extracted.

2) Figure 1, the overlay of synaptic density plots onto the brain are performed for the right hemisphere only, and mirrored to the left. It would be useful to know the reason for this choice: is it commonly assumed that synaptic density is specular in both hemispheres, even knowing that structurally and functionally the brain is asymmetric? (there are multiple references in rodent fMRI literature for this).

3) Figure 2d, I personally struggle to understand what this graph is trying to show.

4) I really appreciate the transparency and clarity of the authors in terms of limitations in their work, one of these being the focus on two synaptic types. Yet, to generalize from two proteins to the entire "synaptome" and its influence on causally shaping whole-brain dynamics is an overstatement.

5) Along the lines of the limitations of this work, the authors clearly state the major issue being the synaptic density analysis performed on N=1 male mouse. First of all, I appreciate keeping a consistent analysis with only utilizing male mice for the fMRI datasets. While I do understand that this article is an initial stepping-stone to create a bridge between micro-scale and macro-functional reorganization (i.e., reproducing all these analyses on a second, third, fourth mouse falls outside the scope of this specific article), it would be informative to investigate how these results relate to the fMRI profiles of female mice.

6) Similarly to #5, commenting on the age differences in the PND80 mouse and the fMRI datasets (were they all the same age?) would help contextualize the results. Sorry if I missed this.

7) These are more technical questions, out of curiosity (therefore optional). The slice thickness of the fMRI data shows quite thick slices (600 um in one direction). Can this bias the results, compared to having thinner slices? How robust are the results with and without gsr? I am also very curious in single-mouse fMRI results: how variable are those from the average utilized?

8) This is personal preference, but I find the color scale chosen to be very hard to interpret and follow, especially in the figures where both ROIs and density are shown, which basically have overlapping colors. Unless this is made on purpose (for which, please make it clearer), it would perhaps be more visually striking to use a different color scheme to categorize ROIs, rather than using a similar shades of blue/greens of the intensity. I understand it can be troublesome to edit all figures at this stage, so this is optional.

Minor:

1) Fig. 2. BOLD signal amplitude doesn't have a y axis range. Are those timeseries all varying within the same range?

2) Throughout the article, results are reported across the anatomically defined brain regions, one of which is the isocortex. However, 'isocortex' includes a lot of highly specialized and diverse types of regions, which I assume may show different behavior in this work. A deeper dive onto what each subregion (Sensory compared to Motor for example) do, would be interesting, especially in the comparison between awake and anesthesia, where I assume differences within and across these regions where found in the fMRI functional connectivity analyses.

3) The integration of datasets from different sources and spatial resolutions necessitates downsampling the high-resolution synapse data. This is a necessary step, but it means that much of the fine-grained spatial detail from the original synaptome maps is partially lost. A brief mention of this as a methodological limitation in the discussion would be appropriate.

4) I believe anesthezia should be spelled with an s.

Reviewer #2: This work builds on previous studies by the authors regarding the characterization of excitatory synapses, in which they used the expression of proteins (PSD95 and SAP102) and the morphology of individual synapses for the identification of synapse types or "synaptomes". The present work analyzes publicly available datasets on synapse density (Zhu et al., 2018) and synapse protein lifetime (Bulovaite et al. 2022) of the identified synapse types, as well as mouse fMRI data (Gutierrez-Barragan et al. 2022) and structural and gene expression data from the Allen Mouse Brain Connectivity Atlas and the Allen Mouse Brain Atlas respectively (and their related publications). Using this approach, this study focuses on the regional heterogeneity of the synapse types and correlates it with features of region-specific fMRI dynamics, and region-specific anatomical and functional connectivity. It further identifies differential roles of synapse types in shaping the structure-function coupling across the different animal states. Overall, this work connects the microscale (synaptic profiles) to the macroscale (inter regional dynamics) and provides several insights into how different synapses shape brain function. Importantly, this work is a very nice example of how open science allows the integration of work done across labs/institutes. Following are my comments on the manuscript, in order of appearance in the text.

Comments

* Fig. 1a and Fig. S2a are the same plot (and the same as Fig. 3d of Zhu et al.). Likewise, Fig. S2b is the same as Fig. 1b (with some missing labels, e.g., although the y-axis label of S2b pools together the colocalized synapse subtype distributions, the delineated boxes correspond to the PSD95- and SAP102-like cases). Consider removing redundant plotting.

* Figure 1 and elsewhere: The chosen color pallets difficult to follow. The colors used to indicate values (e.g., density or correlation in Fig. 1) are very similar to the colors used to indicated the region's ontological structures (Fig 1c-g). This is especially confusing in Fig. 2a,b where values (synapse density) are shown in coronal slices (using a different color pallet from Fig. 1a for synapse density), while the region ontological structure on the right panels of these subplots use very similar colors. In the same figure, time-series features are shown using a different color scheme in panels 1c and 1e/f - slices. For the scatter plots of Fig 2e,f, do the colors indicate regions? Overall, consider adjusting the color pallets for consistency and discriminability.

* Figures S3-S6: Information about the delineated boxes is missing from the figure legends. In the methods section, how was the number of clusters determined when using the agglomerative clustering method (i.e., what was the stopping criterion)?

* S5: Can the authors comment on the SY_LocalGlobal_std_* feature that seems to have the highest correlation? How is this related to the 'extreme Events' (DN_OutlierInclude) feature?

* S6: Can the authors comment on the SC_FluctAnal… feature (with also high correlation value in S3) that seems to have the highest correlation? How is this related to the stationarity of the signal?

* For results shown in Fig. 3, what are the corresponding results for the 2 types of colocalized synapses?

* Fig. 3f. Why did the authors pick the specific regions for the correlation calculations only for the PSD95-long case?

* How are the results for the synapse-informed model for structure-function coupling if the two colocalized types are included as independent variables?

* S8: All three synapse types induce a change in structure-function coupling (though in different ways, with long-lifetime PSD95 and SAP102 showing an increase and short-lifetime PSD95 showing a decrease in anesthetized mice). Can the authors comment on these differences across the different anesthesia?

* "PSD95 and SAP102 synapse density are derived from a single male mouse, raising the concern that these synapse density maps are specific to the individual We therefore use gene expression data from in situ hybridization experiments in the Allen Mouse Brain Atlas to test whether synapses measured in a single mouse are coexpressed with synaptic genes measured across multiple mice". Although the results shown in Fig. 5 are very interesting and provide functional insights into the different synapse types, this analysis does not address the single male mouse concern of the dataset, as it does not validate the results shown in Fig. 1. The dataset of Bulovaite et al., 2022, which includes multiple animals and results correlate with the clustering result of the single-animal data could maybe partially address this concern? I suggest the authors to rephrase this statement to highlight the importance of the results shown in Fig. 5.

* For the results shown in Fig. 5, what are the corresponding results for the 2 types of colocalized synapses? Do they follow the long-lifetime PSD95 and SAP102 profiles?

* Why did the authors use data only from female mice from the study of Bulovaite et al., 2022?

* Do the authors have insights, based on their results, regarding the role of the colocalized long-lifetime PSD95- and SAP102-like synapse types that could be included in the Discussion section? Also, the use of only male mice for the fMRI data could be commented as limitation of both the analyzed datasets and produced data.

Minor comments:

* Consider including line numbers in the manuscript to assist the review process.

* Page 1: refs [1,4,5] do not follow order of appearance (refs 2,3 are not referenced before).

* Fig. 1b. Indicate in the figure legend what the square boxes represent in the plot, as well as the clustering method (as done in the Supplementary Figures). Also, highlight that this and all similar figures are symmetric (or consider showing half of the matrix).

Reviewer #3: This manuscript by Hansen et al. presents an ambitious study that aims to bridge microscale synapse diversity with macroscale brain dynamics using fMRI in the mouse. The integration of large-scale synaptome mapping, functional imaging, and connectivity analysis represents a bold and creative attempt to link levels of brain organization that are rarely studied together. The topic is timely, and the dataset is rich. That said, I have several comments and concerns that I believe should be addressed to strengthen the manuscript.

#1. It is not fully clear whether this work should be primarily considered a methodological resource paper (emphasizing the integration of large datasets and computational approaches) or with novel discoveries to bridge cross-scale brain anatomy and function. At present, the study feels somewhat in-between. A clearer articulation of scope—resource versus discovery-driven—would help readers and clarify the intended contribution.

#2. The reported associations between long-lifetime PSD95 synapses and stationarity, or between SAP102 synapses and high-amplitude events, are intriguing. However, the mechanistic link between synapse architecture and rs-fMRI signal fluctuation remains underdeveloped. For example, it is unclear how specific synaptic distributions translate into region-specific neuronal activity patterns and ultimately into hemodynamic variability. The unresolved challenge of disentangling vascular versus neuronal contributions to rs-fMRI signals further complicates their interpretation. A more explicit discussion of these limitations and how they might be addressed in future work would be valuable.

#3. The authors acknowledge that synapse density, tract-tracing, and fMRI data were derived from different mouse populations. From a rodent fMRI perspective, a key priority would be to establish the reproducibility of the spatial distribution of the >6,000 extracted temporal features across animals. Such evidence would strengthen confidence in the cross-scale association-related discoveries rather than dataset-specific effects.

In summary, this is a bold and inspiring piece of work that pushes boundaries by attempting to link microscale synapse architecture with macroscale brain dynamics.

---

## [Decision Letter · Decision Letter 2]

8 Jan 2026

Dear Dr Misic,

Thank you for your patience while we considered your revised manuscript "Spatial associations between synaptic architecture and regional dynamics in the mouse brain" for publication as a Research Article at PLOS Biology. This revised version of your manuscript has been evaluated by the PLOS Biology editors, the Academic Editor, and the original reviewers.

Based on the reviews, we are likely to accept this manuscript for publication, provided you satisfactorily address the remaining point raised by Reviewer 3. Please also make sure to address the following data and other policy-related requests.

----------------

IMPORTANT: Please ensure that your next revision addresses the following editorial points:

**Title:

-- The current title would potentially work, but we think that it would be helpful to provide some more specificity. Would the following suggestion (or something like it) be acceptable to you?

"Variations in synapse types and density are associated with differences in whole-brain hemodynamics"

**Financial disclosure statement:

-- Please provide all relevant grant numbers and links to the funding agencies in the Financial Disclosure statement in the manuscript details. Please also correct the typo in the word 'acknowledges'.

**Data:

-- Thank you for uploading the underlying data. We also request that you provide the numerical data underlying some of the figure panels, and indicate where these data can be found in the associated figure legends (or just add the location if these data are already present in your repository). If you add an additional supplementary file, please refer to it as "S1 Data" (upload as S1_Data.xlsx). This applies to the following figure panels:

4C

S9B

S10

S11B

S15

**Supplement:

-- We see that you have included the supplemental figures and their legends together in one supplemental file. However, because supplemental files like these are not proofread and are rarely examined by readers, we would prefer if you uploaded the supplemental figures separately, and included their captions/legends at the end of the main text.

**Code availability:

-- Thank you for providing the underlying code in GitHub. However, because Github depositions can be readily changed or deleted, please make a permanent DOI’d copy (e.g. in Zenodo) and provide this URL in the manuscript and Data Availability Statement.

----------------

We expect to receive your revised manuscript within two weeks.

*Published Peer Review History*

*Press*

Sincerely,

Taylor

Taylor Hart, PhD,

Associate Editor

thart@plos.org

PLOS Biology

Reviewer remarks:

Reviewer #1 [Francesca Mandino]: The authors have thoroughly address all reviewers' comments. The manuscript has been greatly improved. I have no further comments.

Reviewer #2: The authors have done an impressive work on addressing the concerns raised. This work now clearly shows a way to correlate the microscale synapse architecture with macroscale brain dynamics and discusses convincingly the limitations/future directions. I have no additional comments.

Reviewer #3: The authors' responses are thoughtful, but a potential conceptual gap remains in the interpretation of resting-state fMRI signals. In particular, the manuscript gives limited consideration to hemodynamic and vascular contributions to rs-fMRI, despite aiming to link synaptic architecture to fMRI dynamics. Several emphasized features of fMRI signal fluctuations in this manuscript are known to be sensitive to vascular and hemodynamic variability, yet the interpretation remains largely neuronal. I am not requesting additional analyses, but a clearer acknowledgment of these factors would help balance the narrative and better define the limits of the proposed synapse-fMRI link.

---

## [Editor Report · Decision Letter 3]

20 Jan 2026

Dear Bratislav,

Thank you for the submission of your revised Research Article "Synapse types are spatially associated with regional hemodynamics in the mouse brain" for publication in PLOS Biology. On behalf of my colleagues and the Academic Editor, Alberto Bacci, I am pleased to say that we can in principle accept your manuscript for publication, provided you address any remaining formatting and reporting issues. These will be detailed in an email you should receive within 2-3 business days from our colleagues in the journal operations team; no action is required from you until then. Please note that we will not be able to formally accept your manuscript and schedule it for publication until you have completed any requested changes.

PRESS

Sincerely,

Taylor

Taylor Hart, PhD,

Associate Editor

PLOS Biology

thart@plos.org